# SliM-LLM: Salience-Driven Mixed-Precision Quantization for Large Language Models

Wei Huang[1]   Haotong Qin[✉ 2]   Yangdong Liu[3]   Yawei Li[2]   Qinshuo Liu[1]
Xianglong Liu[3]   Luca Benini[2]   Michele Magno[2]   Shiming Zhang[✉ 1]   Xiaojuan Qi[✉ 1]

## Abstract

Post-training quantization (PTQ) is an effective technique for compressing large language models (LLMs). However, while uniform-precision quantization is computationally efficient, it often compromises model performance. To address this, we propose SliM-LLM, a salience-driven mixed-precision quantization framework that allocates bit-widths at the group-wise. Our approach leverages the observation that important weights follow a structured distribution and introduces two key components: **1)** *Salience-Determined Bit Allocation* adaptively assigns bit-widths to groups within each layer based on their salience; and **2)** *Salience-Weighted Quantizer Calibration* optimizes quantizer parameters by incorporating element-level salience. With its structured partitioning, SliM-LLM provides a hardware-friendly solution that matches the efficiency of uniform quantization methods while improving accuracy. Experiments show that SliM-LLM achieves superior performance across various LLMs at low bit-widths. For example, a 2-bit quantized LLaMA-7B model reduces memory usage by nearly 6x compared to the floating-point baseline, decreases perplexity by 48% compared to state-of-the-art gradient-free PTQ methods, and maintains GPU inference speed. Additionally, the extended version, SliM-LLM$^+$, which incorporates gradient-based quantization, further reduces perplexity by 35.1%. Our code is available at https://github.com/Aaronhuang-778/SliM-LLM.

[1]The University of Hong Kong [2]ETH Zürich [3]Beihang University. Correspondence to: Haotong Qin, Shiming Zhang, Xiaojuan Qi <haotong.qin@pbl.ee.ethz.ch, beszahng@hku.hk, xjqi@eee.hku.hk>.

*Proceedings of the 42$^{nd}$ International Conference on Machine Learning*, Vancouver, Canada. PMLR 267, 2025. Copyright 2025 by the author(s).

## 1. Introduction

LLMs have demonstrated remarkable performance on various natural language benchmarks (Brown et al., 2020; Hendrycks et al., 2020). Models like LLaMA (Touvron et al., 2023a) and GPT (Brown et al., 2020) have driven progress toward universal language intelligence. Their capabilities have also extended to multi-modal domains (Li et al., 2024b; Achiam et al., 2023; Team et al., 2023; Zhang et al., 2023; Huang et al., 2024b), advancing efforts toward artificial general intelligence (AGI) (Bubeck et al., 2023). However, their high computational and memory demands remain a significant challenge for practical deployment.

To address resource constraints of LLMs, PTQ has emerged as an efficient yet effective compression technique (Dettmers et al., 2022), showing success in quantizing the weights of pre-trained LLMs (Frantar et al., 2022; Lin et al., 2023; Shao et al., 2023; Lee et al., 2023; Chee et al., 2024). As LLMs continue to scale, the demand for more aggressive low-bit compression becomes critical due to limited computational and storage resources in application (Huang et al., 2024a; Tseng et al., 2024). However, significant performance degradation remains a challenge in low-bit scenarios ($\leqslant$ 3-bit). To mitigate this, unstructured mixed-precision quantization (Shang et al., 2023; Huang et al., 2024a; Dettmers et al., 2023) and vector quantization (Chee et al., 2024; Tseng et al., 2024; Egiazarian et al., 2024) methods have been developed to preserve performance. While these approaches have advanced the field, they are often hardware-unfriendly, introducing extra storage requirements such as storing bitmaps or code indices. This creates a bottleneck, limiting further reductions in memory and computational demands during deployments.

This paper presents a **Sal**ience-Driven **M**ixed-Group LLM (**SliM-LLM**) framework, an accurate and inference-efficient PTQ method for LLMs ($\leqslant$ 3-bit). Our approach is grounded in the key observation that *salient or important weights, which are critical to model performance, exhibit a structured distribution, often clustering within certain channels* (see Sec. 3.2.1 and Fig. 3). This insight, largely overlooked by prior research (Frantar et al., 2022), forms the basis for designing SliM-LLM as a structured, hardware-friendly

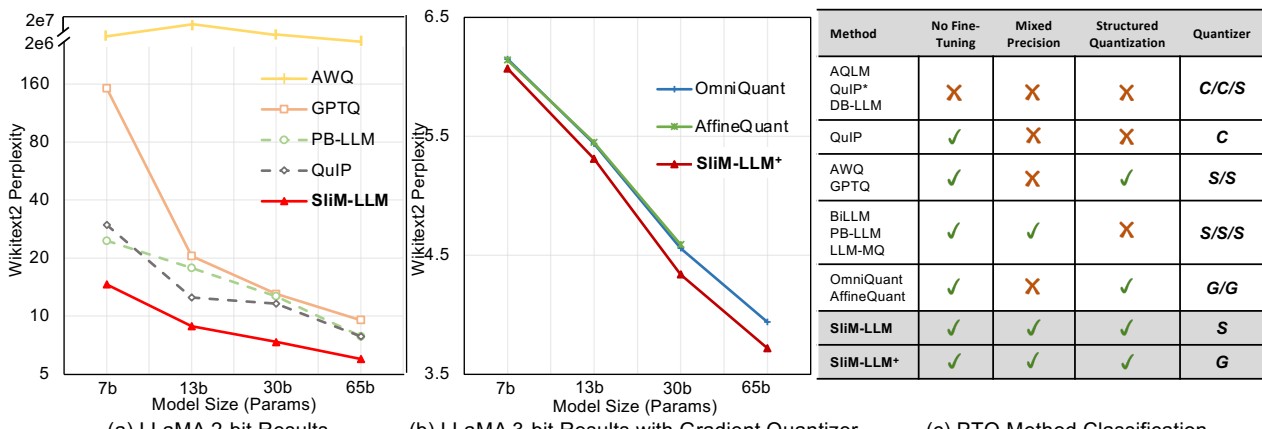

(a) LLaMA 2-bit Results  (b) LLaMA 3-bit Results with Gradient Quantizer  (c) PTQ Method Classification

*Figure 1.* (a) The perplexity (↓) of existing low-bit PTQ methods of LLaMA at 2-bit. Solid-line indicates methods with structured quantization group. (b) Compare PTQ methods with gradient quantizer at 3-bit. (c) Features of current low-bit quantization methods. **C** denotes codebook-based, **S** is statistic-based, and **G** represents gradient-based quantizers.

mixed-precision low-bit method. It preserves performance through two key designs that retain important weights at both the global group and local element levels. First, we develop a novel *Salience-Determined Bit Allocation* (SBA) method, which adaptively assigns bit-widths to each quantization group based on their group-level salience ranking. The allocation strategy is optimized to reduce activation reconstruction errors. By applying higher precision to more important groups and reducing the bit-widths for less critical ones, SBA achieves a low average bit-widths while enhancing the overall performance of LLMs. Next, we introduce the *Salience-Weighted Quantizer Calibration* (SQC), which enhances sensitivity to locally salient weights, ensuring that critical information within groups is preserved. SQC works collaboratively with SBA, exploiting the local and global salience of weights to preserve the performance of LLMs after quantization. Unlike element-wise mixed-precision methods (Shang et al., 2023; Dettmers et al., 2023; Huang et al., 2024a), SliM-LLM is inherently structured, eliminating additional bit or computational overhead while preserving high performance. This is further demonstrated through our deployment of SliM-LLM in an application-level inference tool [1] for LLMs, enabling efficient mixed-precision inference on GPUs with consistently strong performance.

Experiments show that for various LLM families, SliM-LLM surpasses existing training-free PTQ methods on diverse benchmarks, particularly in low-bit scenarios. Using GPTQ as the backbone, SliM-LLM improves the perplexity scores of 2-bit LLaMA-13B and LLaMA2-13B on WikiText2 (Merity et al., 2016) from 20.44 and 28.14 to 8.87 and 9.41, denoting performance improvements of over 56%, respectively. SliM-LLM even outperforms other element-wise mixed-precision PTQ methods, such as PB-LLM (Shang et al., 2023), APTQ (Guan et al., 2024) and LLM-MQ (Li et al., 2024a), in a deployment-friendly manner, showcasing

its superior low-bit accuracy and efficiency. We also integrate SliM-LLM into OmniQuant (Shao et al., 2023) and obtain SliM-LLM[+] through gradient optimization to further improve quantization quality. Moreover, the group-wise mixed-precision strategy can smoothly be adapted to existing quantization-aware training (QAT) (Liu et al., 2023), fine-tuning based (Guo et al., 2023; Liao & Monz, 2024; Dettmers et al., 2024), or codebook-based (Chee et al., 2024; Egiazarian et al., 2024; Tseng et al., 2024) LLMs compression methodologies. This structure of weight salience introduces a new practical view of compression for LLMs.

## 2. Related Work

**Large Language Models** have been significantly developed in diverse natural language processing domains, establishing a prominent paradigm in these fields (Bubeck et al., 2023; Chang et al., 2024; Zhao et al., 2023; Brown et al., 2020; Touvron et al., 2023a). Nevertheless, the exceptional success of LLMs depends on massive parameters and computations, posing significant challenges for deployment in resource-constrained environments. Consequently, research into the compression of LLMs has emerged as a promising field. Existing compression techniques for LLMs primarily include quantization, pruning and distillation (Xu et al., 2023; Ganesh et al., 2021; Frantar et al., 2022; Xiao et al., 2023a; Shao et al., 2023; Chee et al., 2024; Zhu et al., 2023; Frantar & Alistarh, 2023; Huang et al., 2024a; Qin et al., 2024). Low-bit quantization has received notable attention for efficiently reducing the size of the model without changing the structure of the network(Zhu et al., 2023; Zhao et al., 2023; Chang et al., 2024).

**Quantization of LLMs** can generally be categorized into Quantization-Aware Training (QAT) and PTQ. PTQ has emerged as a more practical alternative. Techniques such as LLM.int8()(Liu et al., 2023) and ZeroQuant(Yao et al.,

---
[1] https://github.com/AutoGPTQ/AutoGPTQ

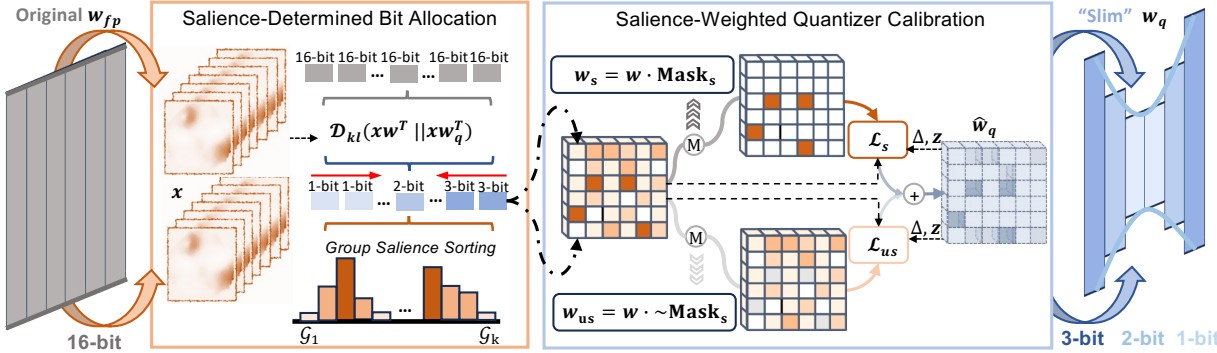

*Figure 2.* Illustration of our proposed SliM-LLM. The *Salience-Determined Bit Allocation* (SBA) optimizes activation-aware structured precision, optimizing the global information distribution in quantization. *Salience-Weighted Quantizer Calibration* (SQC) detects discretely distributed salient weights, enhancing the local important information in LLMs.

2022) introduce block-wise quantization, a cost-effective grouping method that reduces hardware overhead. Further advancements, including AWQ (Lin et al., 2023) and OWQ (Lee et al., 2023), apply scaling transformations to outlier weight channels, preserving their representational capacity. GPTQ (Frantar et al., 2022) minimizes group quantization errors using Hessian-based error compensation (Frantar & Alistarh, 2022), achieving notable performance at 3-bit quantization. OmniQuant (Shao et al., 2023) introduces a learnable scaling quantizer to mitigate quantization errors in an output-aware manner. For ultra-low bit-widths quantization, approaches such as QuIP (Chee et al., 2024), QuIP#(Tseng et al., 2024), and AQLM(Egiazarian et al., 2024) improve 2-bit quantization performance through matrix decomposition with learnable codebooks and fine-tuning. Recent works (Qin et al., 2024; Liao & Monz, 2024; Dettmers et al., 2024; Guo et al., 2023) have further refined quantization techniques by leveraging parameter-efficient fine-tuning (PEFT), enabling enhanced performance through additional parameter learning.

**Mixed-Precision Quantization** leverages the varying importance and redundancy of model parameters by assigning different bit-widths to each component. In traditional visual networks, HAWQ V2 (Dong et al., 2020) and HAWQ V3 (Yao et al., 2021) optimize bit-widths allocation on a layer-wise basis using Hessian analysis and Integer Linear Programming (ILP). Similarly, OMPQ (Ma et al., 2023) employs network orthogonality instead of Hessian for bit-widths optimization. For large language models (LLMs), APTQ (Guan et al., 2024) extends HAWQ's approach by allocating mixed bit-widths to transformer blocks based on Hessian trace, achieving improved accuracy for 3-bit quantization. However, block-wise or layer-wise mixed-precision strategies at 2-bit fail to maintain performance after compression. To address this, recent methods such as SpQR (Dettmers et al., 2023), PB-LLM (Shang et al., 2023), and LLM-MQ (Li et al., 2023) adopt finer-grained

grouping with element-wise mixed-precision for more accurate weight quantization. Despite their improvements, these low-bit approaches rely heavily on fine-grained partitioning, which imposes significant challenges for real hardware deployment and inference speed.

## 3. SliM-LLM

This section introduces a structured mixed-precision quantization method, SliM-LLM, to overcome the accuracy and efficiency bottlenecks of mixed-precision LLMs. We devise two novel strategies, including the use of *Salience-Determined Bit Allocation* (SBA) based on global salience distribution to determine group bit-widths, and *Salience-Weighted Quantizer Calibration* (SQC) to enhance the perception of locally important weight information. We introduce SBA and SQC in Sec. 3.2 and Sec. 3.3, respectively.

### 3.1. Preliminaries

**Quantization Framework.** We first present the general uniform quantization process of LLMs according to common practice (Liu et al., 2023; Shao et al., 2023; Achiam et al., 2023). The quantization process requires mapping float-point weights distributed within the interval $[w_{\min}, w_{\max}]$ to an integer range of $2^N$, where $N$ is the target bit-widths. The quantization function for weight $\boldsymbol{w}_f \in \mathbb{R}^{n \times m}$ follows:

$$\begin{cases} \hat{\boldsymbol{w}}_q = \text{clamp}(\lfloor \frac{\boldsymbol{w}_f}{s} \rceil + z, 0, 2^N - 1), \\ s = \frac{w_{\max} - w_{\min}}{2^N - 1}, z = -\lfloor \frac{w_{\min}}{s} \rceil \end{cases} \quad (1)$$

where $\hat{\boldsymbol{w}}_q$ indicates quantized weight which is integer, $\lfloor \cdot \rceil$ is round operation and $\text{clamp}(\cdot)$ constrains the value within integer range (e.g. $[0, 1, 2, 3]$, $N = 2$). $\Delta$ is scale factor and $z$ is quantization zero point, respectively. When converted

to 1-bit quantization, the calculation follows:

$$\hat{\boldsymbol{w}}_b = \text{sign}(\boldsymbol{w}_f), \alpha = \frac{1}{l}||\boldsymbol{w}_f||_{\ell 1}$$
$$\text{sign}(w) = \begin{cases} 1 & \text{if } w \geq 0, \\ -1 & \text{others.} \end{cases} \quad (2)$$

where $\hat{\boldsymbol{w}}_b$ is binary result. $\alpha$ denots binarization scales and $l$ is the number of elements in weight (Qin et al., 2023), used for dequantization through $\alpha\hat{\boldsymbol{w}}_b$. We can formalize the per-layer loss in PTQ, following the common practice (Nagel et al., 2020; Frantar et al., 2022):

$$\mathcal{L}(\hat{\boldsymbol{w}}_f) = ||\boldsymbol{x}\boldsymbol{w}_f^\top - \boldsymbol{x}\hat{\boldsymbol{w}}_f^\top||^2 \approx \text{tr}((\hat{\boldsymbol{w}}_f - \boldsymbol{w})\boldsymbol{H}(\hat{\boldsymbol{w}}_f - \boldsymbol{w})^\top) \quad (3)$$

where $\boldsymbol{x} \in \mathbb{R}^{t \times m}$ denotes the input vectors from calibration dataset, $\hat{\boldsymbol{w}}_f \in \mathbb{R}^{n \times m}$ is dequantized weight from quantization result in Eq. (1) or Eq. (2), and $\boldsymbol{H} = \frac{1}{P}\sum_{k=1}^{P}\boldsymbol{x}^{[k]^\top}\boldsymbol{x}^{[k]}$ is proxy Hessian matrix by Levenberg-Marquardt approximation (Marquardt, 1963; Frantar & Alistarh, 2022) from a set of input activations.

**Parameter Salience.** In LLMs, the importance of each element in the weight matrix is various (Dettmers et al., 2023; Frantar & Alistarh, 2023). According to Eq. (3), quantizing different elements causes different impacts on the model's output loss. Elements that significantly influence the loss are termed salient weights. Consequently, we follow the SparseGPT (Frantar & Alistarh, 2023) to define the salience of each element as:

**Definition 3.1.** In the quadratic approximation of the loss as expressed in Eq. (3), we give the Hessian matrix $H \in \mathbb{R}^{m \times m}$ generated by $\frac{1}{P}\sum_{k=1}^{P}\boldsymbol{x}^{[k]^\top}\boldsymbol{x}^{[k]}$ for a weight matrix, the removal of the element at $(i, j)$ induces an error $\delta_{i,j} = \frac{w_{i,j}^2}{[\boldsymbol{H}^{-1}]_{j,j}^2}$ to the output matrix for linear projection in LLMs.

where $[\boldsymbol{H}^{-1}]_{jj}$ denotes the $j^{th}$ diagonal entry for the inverse Hessian, and $\boldsymbol{H}^{-1}$ can be efficiently calculated through Cholesky decomposition (Krishnamoorthy & Menon, 2013). According to Definition. 3.1, we map the elimination error $\delta_{ij}$ to the salience measure of each weight element in LLMs, representing the impact of different weights on the output loss and the language capabilities, which also leads the generation of mixed-precision quantization strategies (Dettmers et al., 2023; Shang et al., 2023; Huang et al., 2024a; Li et al., 2024a) for LLMs. However, existing mixed-precision solutions require the discrete allocation of bit-widths across the entire weight matrix, which imposes a significant burden on hardware computations, thereby affecting the inference efficiency.

### 3.2. Salience-Determined Bit Allocation

We reveal the spatial clustering of weight salience, which inspires our proposed concept of group-wise mixed-precision quantization for LLMs, and then introduce the *Salience-*

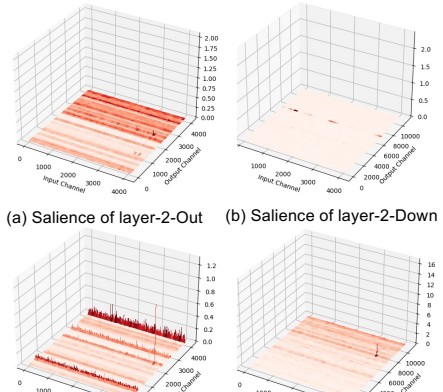

(a) Salience of layer-2-Out  (b) Salience of layer-2-Down

(c) Salience of layer-10-Out  (d) Salience of layer-10-Down

*Figure 3.* Salience weight distribution in layer-2 and layer-10 of LLaMA-7B.

*Determined Bit Allocation* (SBA) technique to allocate the optimal precision to each group.

#### 3.2.1. SPATIAL DISTRIBUTION OF GLOBAL SALIENCE

We first conduct an empirical investigation into the weight salience distribution. The results reveal that certain channels exhibit higher salience and show tendencies to spatial clustering. As illustrated in Fig. 3, salient clustering is identified around the $2100^{th}$, $3218^{th}$ and $3853^{rd}$ channels within the $2^{nd}$ layer's attention projection of the LLaMA-7B model. A similar structured pattern is observed in other layers. Also, clustered salience is detected in other layers (as shown in Fig. 3). More examples are provided in Appendix G.

Then, we analyze the underlying reasons for this phenomenon. According to Definition 3.1, the salience of weights is proportional to the magnitude of the weights and the trace of the Hessian matrix, which can be approximated by the product of input activations $\boldsymbol{x}^\top\boldsymbol{x}$. In LLMs, activations exhibit extreme outlier channels, while the numerical differences in weights are relatively slight (Xiao et al., 2023a; Nrusimha et al., 2024). Therefore, we propose an analysis of how the outlier channels in activations influence the distribution of weight salience:

*Theorem* 1. Given the input calibration activation $\boldsymbol{x} \in \mathbb{R}^{t \times m}$ with an outlier channel $\boldsymbol{x}_{:,p}^* \gg \boldsymbol{x}_{:,j}, \forall j \in [0, m], j \neq p$ at the position of channel-$p$. The trace elements of $\boldsymbol{H} = \boldsymbol{x}^\top\boldsymbol{x}$ will show great outlier value at $(p, p)$, where $\boldsymbol{H}_{p,p} \gg \boldsymbol{H}_{j,j}, \forall j \in [0, m], j \neq p$, as $\boldsymbol{H}_{p,p}$ is produced by $[\boldsymbol{x}_{:,p}^{*\top}\boldsymbol{x}_{:,p}^*] = \sum_{i=0}^{t} x_{i,p}^{*2}$, which further leads to the parameter salience larger at the $p^{th}$ channel of weight, where $\delta_{:,p} > \delta_{:,k}, \delta_{:,k} = \frac{w_{:,k}^2}{[\boldsymbol{H}^{-1}]_{k,k}^2}, \forall k \in [0, t], k \neq p$.

Theorem 1 elucidates the influence of outlier activation on the distribution of channel salient weights (detailed proof in the Appendix G.1). Furthermore, recent research indicates that outlier channels in LLMs activations consistently appear

in fixed yet clustered patterns (Nrusimha et al., 2024). According to Theorem 1, these consistently occurring anomalous activations result in the distribution of salient weights, as depicted in Fig. 3. Then, during group-wise quantization, the average salience of each group shows different features.

Meanwhile, previous unstructured mixed-precision, incurred additional storage requirements and computational overheads, affecting the real-time inference. However, the strong spatial structured characteristics observed in the salient of weights in this section strongly inspire us to first develop a group-wise mixed-precision strategy within the weight matrix while maintaining inference efficiency. Therefore, we aim to allocate bit-widths based on intra-group salient disparities, which not only enhances quantization accuracy but also ensures the inference efficiency of LLMs with structured bit-widths saving and dequantization.

### 3.2.2. GROUP-WISE BIT ALLOCATION

To allocate optimal bit-widths to each group, we introduce a *Salience-Determined Bit Allocation* (SBA) technique for mixed-precision LLMs, as depicted in Fig. 2. This technique, predicated on the differences in group salience, determines the optimal bit-widths allocation for different groups by minimizing the distance of information entropy with the original weight output. Specifically, we first utilize the average salience as the importance indicator for each weight group and rank them accordingly. Then, the proposed SBA optimizes the bit allocation following:

$$
\begin{aligned}
\text{Objective}: &\arg\min \mathcal{D}_{kl}\left(\boldsymbol{x}\boldsymbol{w}_f^\top \,||\, \boldsymbol{x}(\hat{\boldsymbol{w}}_{sba})^\top\right), \\
&\text{where } \hat{\boldsymbol{w}}_{sba} = [\hat{\boldsymbol{w}}_{0,b_0}, \hat{\boldsymbol{w}}_{1,b_1}...\hat{\boldsymbol{w}}_{k-1,b_{k-1}}, \hat{\boldsymbol{w}}_{k,b_k}] \\
\text{Constrain}: &|\mathcal{G}_{N-1}| = |\mathcal{G}_{N+1}|, \\
&\text{where } \mathcal{G}_{N-1} = \{b_i | b_i = N - 1\}, \\
&\mathcal{G}_{N+1} = \{b_j | b_j = N + 1\}
\end{aligned}
$$
(4)

where $\mathcal{D}_{kl}(\cdot||\cdot)$ denotes the Kullback-Leibler (KL) divergence between two outputs, $\hat{\boldsymbol{w}}_f^{sba}$ generally represents the de-quantization results of weight, employing group-wise mixed-precision designated as $[\hat{\boldsymbol{w}}_{0,b_0}, \hat{\boldsymbol{w}}_{1,b_1}...\hat{\boldsymbol{w}}_{k-1,b_{k-1}}, \hat{\boldsymbol{w}}_{k,b_k}]$, where $b_i$ represents the bit-widths for the $i^{th}$ group and $\mathcal{G}$ is a set of groups with the same bit-width, $N$ is the targeted average bit-width. We apply a compensation constraints strategy to maintain a consistent average bits for our SBA. For example, in 2-bit quantization, the more salient groups are quantized to 3-bit. To offset the additional bits, we quantize an equal number of groups with the lower salience to 1-bit ($|\mathcal{G}_{N-1}| = |\mathcal{G}_{N+1}|$), while the remaining groups are set to 2-bit.

We utilize an effective double-pointer search (more detailed examples in Appendix C) to optimize our objective in Eq. (4). When the weight output channel size is $m$ and group

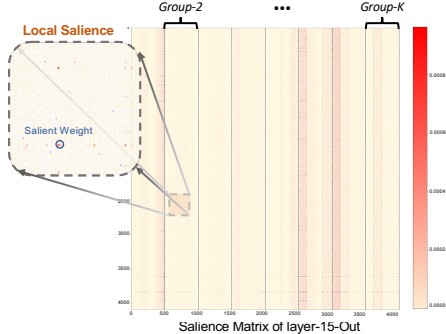

Figure 4. Local salience distribution of the $10^{th}$ MHA output layer.

size is 128, $k = \frac{m}{128}$, the search region for weight is limited to $[0, \frac{k}{2}]$, which is highly efficient with limited searching space, *e.g.*, only 16 iterations are needed in LLaMA-7B. We also provide detailed searching error examples in Appendix C. Notably, SBA diverges from traditional quantization with mean squared error (MSE) in Eq. (3) by instead utilizing the KL divergence as its measure of loss. Compared to using mean squared error (MSE) for weights, SBA utilizes the KL divergence of block outputs as a precision allocation metric, aiming to align the distribution of the LLM's output activation matrix with that of the quantized activation. This approach improves the model's information representation under low-bit quantization, enabling more effective bit-widths allocation. While HAWQ v2 (Dong et al., 2019) employs Integer Linear Programming (ILP) to allocate bit-widths for layers, its method can be adapted to group-wise targets. Detailed comparisons between SBA and ILP are provided in Section 4.2.

### 3.3. Salience-Weighted Quantizer Calibration

In addition to the global group-wise distribution of salience, we notice that salience within the group still shows local differences in discrete distribution. Common existing quantizers apply uniform consideration across all weights to minimize the effect (error) of quantization, lacking the capability to perceive differences in local salience. Therefore, in this section, we introduce a *Salience-Weighted Quantizer Calibration* (SQC) to enhance the information of significant weights within the group by amplifying the quantizer awareness of salient weight.

### 3.3.1. DISCRETE DISTRIBUTION OF LOCAL SALIENCE

In the aforementioned section, we group-wisely allocate the bit-widths for each group based on the global salience. To maintain the efficiency of quantized inference, we employ a commonly used sequential structured grouping (Frantar et al., 2022; Lin et al., 2023; Shao et al., 2023). However, this group-wise mixed-precision also leads to differences in salience among the various elements within the same group. Specifically, as the salience distribution in Fig. 4, within

the $10^{th}$ attention output layer of LLaMA-7B, a subset of sparse weights within the comparatively less salient Group-2 (Fig. 4) still maintains a high level of importance. In LLMs, a small number of weight elements with outliers affect the local distribution of salience. These discrete weights typically account for only approximately 1% within the group but play a crucial role in the modeling capability of LLMs.

Vanilla quantizers struggle to represent local salient information as they only consider the mean error of all elements within a group. During group-wise quantization (Eq. 1), non-salient weights often dominate, degrading critical information and negatively impacting LLM performance.

### 3.3.2. SALIENCE-WEIGHTED QUANTIZER

To prevent the degradation of local salient information in each group, we propose the *Salience-Weighted Quantizer Calibration* (SQC), which enhances the expression of salient weights through local salience awareness, thereby reducing the quantization error of these significant elements and improving the compressed performance of LLMs.

Based on a common observation (Dettmers et al., 2023; Huang et al., 2024a), the proportion of relatively salient weights in each group is only 1-5%. Therefore, we employ the 3-$\sigma$ rule for a mask to select the salience part $(\boldsymbol{w} < (\mu - 3\sigma) \cup \boldsymbol{w} > (\mu + 3\sigma))$ in each group (Fig. 2), which accounts for about 1% elements. After the selection, we get $\boldsymbol{w}_i = \boldsymbol{w}_i^s \cup \boldsymbol{w}_i^{us}$, where $\boldsymbol{w}_i^s$ is the salient part and $\boldsymbol{w}_i^{us}$ represents the non-salient elements within group $i$. To effectively keep the information of local salient weights, SQC first introduces the calibration parameter $\tau$ to the SQC quantizer, liberating the perception interval during quantization. Then we define the local salience awareness loss of the SQC quantizer through calibration:

$$\begin{aligned} \operatorname*{argmin}_{\tau} \ &||\boldsymbol{w}_i^s - \tau \cdot s\{\mathcal{Q}(\boldsymbol{w}_i^s, \tau \cdot s, \tau \cdot z) - \tau \cdot z\}||_2^2 + \\ &||\boldsymbol{w}_i^{us} - \tau \cdot s(\mathcal{Q}(\boldsymbol{w}_i^{us}, \tau \cdot s, \tau \cdot z) - \tau \cdot z)||_2^2 \end{aligned} \quad (5)$$

where $\mathcal{Q}(\cdot)$ denotes the quantization process in Eq. (1), $||\cdot||_2^2$ represents the $\ell_2$ loss, aligned with Eq. (3). $\boldsymbol{w}_i^s$ and $\boldsymbol{w}_i^{us}$ denotes the salient and less salient part of group $i$, respectively, generated from a mask operation. In Eq. (5), $\tau$ expands the solution space of $s$ and $z$, flexibly adjusts $s$ and $z$ to search the optimal loss under $\tau^*$, without bringing additional parameters, as $\boldsymbol{w}_i^s$ and $\boldsymbol{w}_i^{us}$ share the same quantizer. The search space for $\tau$ by linearly dividing the interval [1-$\lambda$, 1+$\lambda$] into $2n$ candidates. We empirically set $\lambda$ at 0.1 and $n$ at 50 to achieve a balance between efficiency and accuracy.

Compared to traditional quantizer calibration methods, SQC effectively mitigates the degradation of intra-group local salient weights caused by general average loss by enhancing the loss sensitivity to salient elements during the calibration (more experiments are detailed in Appendix E). More-

over, the SQC process allows $\boldsymbol{w}_i^s$ and $\boldsymbol{w}_i^{us}$ to share a set of parameters $\tau^*s$ and $\tau^*z$, eliminating the need to differentiate intra-group weights during storage and inference. This facilitates straightforward group-wise dequantization calculations, thereby avoiding the hardware overhead associated with element-wise bitmap and unstructured grouping. SQC and SBA capture both local salient weight information within groups and global weight combinations across groups, effectively preserving critical information and maintaining LLM performance at extremely low bit-widths.

### 3.4. Implementation Pipeline of SliM-LLM

We integrate our mixed-precision framework into advanced PTQ methods, such as GPTQ (Frantar et al., 2022) and OmniQuant (Shao et al., 2023), all of which are inference-friendly with group-wise quantization. We primarily integrate SBA and SQC into GPTQ to get SliM-LLM. For SliM-LLM$^+$, the SBA is plugged into OmniQuant with a learnable quantizer. The plugging pipeline of SliM-LLM is provided in Algorithm 1 (line 4 and line 9), detailed functions are shown in Algorithm 2.

## 4. Experiments

We evaluated SliM-LLM and SliM-LLM$^+$ under weight-only conditions, focusing on 2/3-bit precisions. Per-channel group quantization is utilized in our framework with $groupsize = 128$ in experiments. Since no back-propagation in SliM-LLM, the quantization is carried out on a single NVIDIA A800 GPU. For SliM-LLM$^+$, we employ the AdamW optimizer, following OmniQuant (Shao et al., 2023), which is also feasible on a single A800. We randomly select 128 samples from WikiText2 (Merity et al., 2016) as calibration data, each with 2048 tokens.

**Models and Evaluation.** To comprehensively demonstrate the low-bit performance advantages of SliM-LLM and SliM-LLM$^+$, we conduct experiments across OPT (Zhang et al., 2022), LLaMA (Touvron et al., 2023a), LLaMA-2 (Touvron et al., 2023b) and LLaMA-3. We employ the perplexity as our evaluation metric, which is widely recognized as a stable measure of language generation capabilities (Frantar et al., 2022; Lin et al., 2023; Huang et al., 2024a; Shang et al., 2023; Shao et al., 2023; Chee et al., 2024; Egiazarian et al., 2024; Huang et al., 2024b), particularly in compression scenarios. Experiments are carried out on the WikiText2 (Merity et al., 2016) and C4 (Raffel et al., 2020)datasets. Furthermore, to assess the practical application capabilities of quantized LLMs, we also evaluate on zero-shot benchmarks.

### 4.1. Main Results

We show experiments within the LLaMA family in this section and results for the OPT models are available in

*Table 1.* Quantization results of LLaMA family with statistic quantizer. We report the WikiText2 perplexity in this table, C4 results are shown in Appendix H. '-' denotes that the selected works did not give the results on listed models or the codes.

| #W PPL↓ | Method | 1-7B | 1-13B | 1-30B | 1-65B | 2-7B | 2-13B | 2-70B | 3-8B | 3-70B |
|---|---|---|---|---|---|---|---|---|---|---|
| 16-bit | - | 5.68 | 5.09 | 4.10 | 3.53 | 5.47 | 4.88 | 3.31 | 5.75 | 2.9 |
| 3-bit | APTQ | 6.76 | - | - | - | - | - | - | - | - |
| | LLM-MQ | - | - | - | - | - | 8.54 | - | - | - |
| | RTN | 7.01 | 5.88 | 4.87 | 4.24 | 6.66 | 5.51 | 3.97 | 27.91 | 11.84 |
| | AWQ | 6.46 | 5.51 | 4.63 | 3.99 | 6.24 | 5.32 | - | 8.22 | 4.81 |
| | GPTQ | 6.55 | 5.62 | 4.80 | 4.17 | 6.29 | 5.42 | 3.85 | 8.19 | 5.22 |
| | **SliM-LLM** | **6.40** | **5.48** | **4.61** | **3.99** | **6.24** | **5.26** | **3.67** | **7.16** | **4.08** |
| 2-bit | LLM-MQ | - | - | - | - | - | 12.17 | - | - | - |
| | RTN | 1.9e3 | 781.20 | 68.04 | 15.08 | 4.2e3 | 122.08 | 27.27 | 1.9e3 | 4.6e5 |
| | AWQ | 2.6e5 | 2.8e5 | 2.4e5 | 7.4e4 | 2.2e5 | 1.2e5 | - | 1.7e6 | 1.7e6 |
| | GPTQ | 152.31 | 20.44 | 13.01 | 9.51 | 60.45 | 28.14 | 8.78 | 210.00 | 11.90 |
| | QuIP | 29.74 | 12.48 | 11.57 | 7.83 | 39.73 | 13.48 | 6.64 | 84.97 | 13.03 |
| | PB-LLM | 24.61 | 17.73 | 12.65 | 7.85 | 25.37 | 49.81 | NAN | 44.12 | 11.68 |
| | **SliM-LLM** | **14.58** | **8.87** | **7.33** | **5.90** | **16.01** | **9.41** | **6.28** | **39.66** | **9.46** |

*Table 2.* Quantization results of LLaMA-1 and LLaMA-2 models with learnable quantizer. We report the WikiText2 perplexity in this Table, C4 results are shown in Appendix H.

| #W PPL↓ | Method | 1-7B | 1-13B | 1-30B | 1-65B | 2-7B | 2-13B | 2-70B |
|---|---|---|---|---|---|---|---|---|
| 16-bit | - | 5.68 | 5.09 | 4.10 | 3.53 | 5.47 | 4.88 | 3.31 |
| 3-bit | OmniQuant | 6.15 | 5.44 | 4.56 | 3.94 | 6.03 | 5.28 | 3.78 |
| | AffineQuant | 6.14 | 5.45 | 4.59 | - | 6.08 | 5.28 | - |
| | **SliM-LLM$^+$** | **6.07** | **5.37** | **4.34** | **3.72** | **5.94** | **5.11** | **3.35** |
| 2-bit | OmniQuant | 9.72 | 7.93 | 7.12 | 5.95 | 11.06 | 8.26 | 6.55 |
| | AffineQuant | 13.51 | 7.22 | 6.49 | - | 10.87 | 7.64 | - |
| | **SliM-LLM$^+$** | **9.68** | **7.17** | **6.41** | **5.74** | **10.87** | **7.59** | **6.44** |

**Appendix H.** For language generation tasks, as depicted in Tab. 1, SliM-LLM markedly outperforms its backbone GPTQ, particularly under the 2-bit. Specifically, on LLaMA-7B, SliM-LLM achieves a 90% decrease in perplexity, while on LLaMA-3-8B, it improves by 81%. In comparison with the element-wise mixed-precision PB-LLM and the codebook-based QuIP method, SliM-LLM further reduces the perplexity by 41%~51%. As shown in Tab. 1, the performance of SliM-LLM$^+$ is still ahead compared to Omni-Quant and AffineQuant. We also provide dialogue examples of 2-bit instruction fine-tuning Vicuna-13B (Chiang et al., 2023) and LLaMA-13B in Appendix I. Our method demonstrates zero-shot advantages at 2-bit, as shown in Tab. 3, where SliM-LLM and SliM-LLM$^+$ outperform other methods. For example, compared to GPTQ and OmniQuant, our approach improves LLaMA-7B performance by 4.19% and 1.91% on average. On LLaMA-65B, 2-bit SliM-LLM and SliM-LLM$^+$ achieve accuracy within 6% of FP16. To further showcase the general performance of SliM-LLM, we also compare the low-bit results on multi-modal models (Tab. 15), where SLiM-LLM presents leading accuracy on 4 benchmarks.

### 4.2. Ablation Results

**Ablation of SBA and SQC.** We conduct a detailed ablation study to illustrate the benefits of bit-widths allocation and

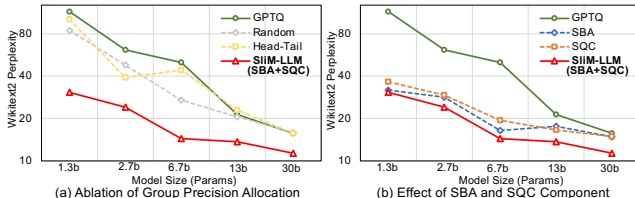

*Figure 5.* Ablation results on OPT models. Random refers to randomly selecting lower- and higher-bit groups, while head-tail assigns lower-bit precision to the head groups and higher-bit precision to an equal number of tail groups in the original sequence.

the impact of each component. Fig. 5(a) compares three strategies for allocating bit-widths across groups, including random allocation, head-tail allocation by spatial order, and our proposed SBA. When the average bit-widths remains constant, random and head-tail mixed-precision allocation prove ineffective and even result in performance degradation, as shown in Fig. 5(a). In contrast, SBA consistently delivers significant improvements in post-quantization performance, validating the efficacy of our mixed-precision approach. Fig. 5(b) presents the ablation effects of SBA and SQC, demonstrating that both methods, based on the perception of global and local salience, enhance quantization performance.

**Compare of SBA and ILP.** We compare the performance

*Table 3.* Performance comparisons of different quantization methods for zero-shot tasks.

| Model / Acc↑ | #W | Method | PIQA | ARC-e | ARC-c | BoolQ | HellaSwag | Winogrande | Avg. |
|---|---|---|---|---|---|---|---|---|---|
| LLaMA-7B | 16-bit | - | 77.47 | 52.48 | 41.46 | 73.08 | 73.00 | 67.07 | 64.09 |
| | 2-bit | GPTQ | 55.49 | 31.02 | 22.17 | 53.49 | 33.84 | 41.91 | 39.65 |
| | 2-bit | AWQ | 47.78 | 28.77 | 21.31 | 31.19 | 24.47 | 40.03 | 32.26 |
| | 2-bit | **SliM-LLM** | **57.83** | **33.46** | **25.09** | **56.05** | **36.70** | **52.64** | **43.84** |
| | 2-bit | OmniQuant | 63.63 | 43.91 | 27.32 | 58.02 | 48.78 | 52.97 | 49.11 |
| | 2-bit | **SliM-LLM$^+$** | **64.96** | **45.66** | **28.67** | **64.59** | **48.86** | **53.35** | **51.02** |
| LLaMA-13B | 16-bit | - | 79.10 | 59.89 | 44.45 | 68.01 | 76.21 | 70.31 | 66.33 |
| | 2-bit | GPTQ | 70.37 | 47.74 | 35.88 | 51.57 | 61.39 | 60.84 | 54.63 |
| | 2-bit | AWQ | 49.23 | 30.01 | 29.49 | 30.88 | 26.72 | 46.30 | 35.44 |
| | 2-bit | **SliM-LLM** | **73.19** | **47.95** | **36.27** | **55.92** | **63.04** | **61.79** | **56.36** |
| | 2-bit | OmniQuant | 73.14 | 49.38 | 36.93 | 63.34 | 62.19 | 61.77 | 57.64 |
| | 2-bit | **SliM-LLM$^+$** | **74.15** | **50.26** | **37.04** | **64.31** | **63.57** | **63.11** | **58.74** |
| LLaMA-30B | 16-bit | - | 80.08 | 58.92 | 45.47 | 68.44 | 79.21 | 72.53 | 67.44 |
| | 2-bit | GPTQ | 71.92 | 48.27 | 36.20 | 61.27 | 65.76 | 63.11 | 57.76 |
| | 2-bit | AWQ | 49.17 | 28.56 | 25.97 | 34.73 | 24.97 | 46.99 | 35.07 |
| | 2-bit | **SliM-LLM** | **75.52** | **51.29** | **39.29** | **62.01** | **66.10** | **64.07** | **59.71** |
| | 2-bit | OmniQuant | 76.23 | 53.23 | 39.52 | 63.34 | 65.57 | 64.82 | 60.22 |
| | 2-bit | **SliM-LLM$^+$** | **76.31** | **54.07** | **39.79** | **63.35** | **67.14** | **64.93** | **60.91** |
| LLaMA-65B | 16-bit | - | 80.79 | 58.71 | 46.24 | 82.29 | 80.72 | 77.50 | 71.04 |
| | 2-bit | GPTQ | 76.16 | 52.48 | 40.14 | 77.23 | 71.96 | 70.22 | 64.70 |
| | 2-bit | **SliM-LLM** | **77.09** | **53.72** | **40.25** | **77.51** | **72.05** | **70.91** | **65.26** |
| | 2-bit | OmniQuant | 77.78 | 53.71 | 40.90 | 78.04 | 74.55 | 68.85 | 65.64 |
| | 2-bit | **SliM-LLM$^+$** | **78.06** | **53.90** | **41.18** | **78.33** | **75.59** | **69.99** | **66.18** |

*Table 4.* WikiText2↓ performance of SBA and ILP on LLaMA.

| Method | #W | 7B | 13B | 30B | 65B |
|---|---|---|---|---|---|
| ILP | 2-bit | 17.55 | 9.51 | 9.27 | 7.46 |
| **SBA** | 2-bit | **14.58** | **8.87** | **7.33** | **5.90** |

*Table 5.* Deployment results of GPTQ and Slim-LLM on GPU. Group size is set to 128.

| #W | LLaMA-* | 1-7B | | | | 1-13B | | | |
|---|---|---|---|---|---|---|---|---|---|
| | | WM | RM | PPL↓ | Token/s | WM | RM | PPL↓ | Token/s |
| FP16 | - | 12.6G | 14.4G | 5.68 | 69.2 | 24.3G | 27.1G | 5.09 | 52.5 |
| 3-bit | GPTQ | 3.2G | 5.1G | 6.55 | 83.4 | 5.8G | 8.7G | 5.62 | 57.6 |
| | **SliM-LLM** | 3.2G | 5.2G | **6.40** | 79.1 | 5.8G | 8.8G | **5.48** | 48.5 |
| 2-bit | GPTQ | 2.2G | 4.4G | 152.31 | 83.9 | 4.0G | 7.6G | 20.44 | 92.6 |
| | **SliM-LLM** | 2.3G | 4.4G | **14.58** | 61.2 | 4.1G | 7.8G | **8.87** | 73.7 |

between the ILP model in HAWQ v2 (Dong et al., 2019) and SBA on the LLaMA model. Tab. 4 shows that SBA achieves comprehensive performance superiority on LLaMA. We observed that under a 2-bit scenario, ILP ensures an equal number of 1-bit and 3-bit groups within the search space {1-bit, 2-bit, 3-bit}. The advantage of ILP lies in a broader selection range for target bit-widths, but under commonly used fixed integer bit-widths (e.g. 2-bit, 3-bit), SBA's double-pointer search strategy based on output feature KL proposed by SBA can achieve a more optimal matching strategy.

### 4.3. Efficient Inference on Device

We utilize the open-source AutoGPTQ to extend CUDA kernel supporting experimental mixed-precision inference, with detailed process in Appendix B.2. As shown in Tab. 5,

we evaluate the deployment performance of LLaMA-7/13B and LLaMA-2-7B under 2/3-bit settings. The results indicate that our mixed-precision approach maintains a good compression rate on GPUs and significantly enhances model accuracy, only with a slight decrease in inference speed on the A800. We also deploy the LLaMA-2-70B in Tab. 14. Since current 1-bit operations lack well hardware support, additional consumption of storage and computation is required on device. There remains considerable scope for optimization in mixed-precision computing, and we aim to further improve this in future work.

## 5. Conclusion

In this work, we propose **SliM-LLM**, a group-wise mixed-precision PTQ framework for LLMs, designed to optimize performance with low-bit weights while maintaining deployment efficiency. The core of SliM-LLM is the Salience-Determined Bit Allocation, which dynamically assigns bit widths to preserve global salience information. Additionally, the Salience-Weighted Quantizer Calibration enhances local information retention, reducing the impact of quantization on locally salient weights. Experimental results demonstrate that SliM-LLM significantly improves accuracy across various LLMs while ensuring inference efficiency. Overall, SliM-LLM is a versatile solution that integrates seamlessly with existing quantization frameworks, enabling practical deployment in resource-constrained environments.

## Acknowledgments

The authors acknowledge the Innovation and Technology Fund from Innovation and Technology Commission of Hong Kong SAR Government (Mainland-Hong Kong Joint Funding Scheme MHP/053/21), Shenzhen-Hong Kong-Macau Technology Research Programme (SGDX20210823103537034) from Shenzhen Science and Technology Innovation Commission, and Seed Funding for Strategic Interdisciplinary Research Scheme from the University of Hong Kong (HKU) for supporting this work. This work has also been supported by Hong Kong Research Grant Council - Early Career Scheme (Grant No. 27209621), General Research Fund Scheme (Grant No. 17202422, 17212923), the Swiss National Science Foundation (SNSF) project 200021E_219943 Neuromorphic Attention Models for Event Data (NAMED)

## Impact Statement

This paper introduces a mixed-precision technique to achieve accurate and efficient low-bit weight quantization for large language models (LLMs). This approach makes LLMs more efficient and accessible, potentially extending their pervasive impact. From a positive perspective, quantization makes the use of LLMs easier, benefiting a broader audience, particularly those in lower-income groups. It reduces the cost and hardware barriers to deploying LLMs and promotes edge inference of these models (mitigating the risk of privacy data breaches), contributing to societal productivity. On the downside, LLMs could be exploited by malicious users to generate and spread false information. Quantization does not prevent the inherent negative impacts of LLMs, nor does it exacerbate them.

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

## A. Limitations

Though the mixed-precision framework significantly improves the quantization performance of LLMs, the current out-of-the-box deployment tools still cannot well support efficient mixed-precision computing. Meanwhile, the support for 1/2/3-bit inference on GPUs remains limited, which affects the inferencing advantages of low-bit models. We believe there is significant room for improvement in the hardware efficiency of mixed-precision LLMs in the future.

## B. SliM-LLM Implementation

### B.1. Detailed Implementation

In this section, we present the specific implementation details of SliM-LLM, which utilizes GPTQ (Frantar et al., 2022) as its backbone for mixed-precision quantization and incorporates both SBA and SQC. SliM-LLM$^+$ is consistent with SliM-LLM in SBA computations but does not include the SQC component, instead retaining learnable weight clipping (LWC) approach in OmniQuant (Shao et al., 2023) for gradient optimization.

Algorithm 2 primarily encompasses the core details of both SBA and SQC. In SBA, the importance of each group is determined by sorting the average salience of groups, followed by a bi-pointer search that increases the number of $(N-1)$-bit and $(N+1)$-bit groups to maintain their quantity equilibrium. The optimization function then utilizes the KL divergence from Eq. (4) to determine the optimal mixed-precision ratio. SQC, on the other hand, enhances its information by amplifying the quantization error of unstructured weight groups. When the last two parameters, scale and zero point, in the fakequant($\cdot$) function are omitted, the default values from Eq. (1) are used.

### B.2. Mixed Bit Storage and Computing

We developed a framework for storage and inference deployment supporting mixed-precision quantization based on AutoGPTQ. The deployment process is as follows. After completing mixed-precision quantization with SliM-LLM, it outputs scales, zeros, and group-wise bit-widths generated during the quantization process to identify the quantization parameters and precision of each group in the Linear Projection weights. AutoGPTQ then packs the weights and zeros into integer-compressed representations (denoted by $\hat{w}_{\text{int}}$ and $\hat{z}_{\text{int}}$ respectively) based on the precision of different groups, significantly reducing storage and operational bit-width. After the quantized weights are packed, AutoGPTQ loads the model onto the GPU, where the mixed precision quantization kernel on the GPU performs dequantization on the weights and zeros of different groups and calculation with input activation, ultimately producing the final output.

In the mixed-precision deployment of AutoGPTQ, the weight memory layout is organized by group, with each group sharing the same precision, which is shown in Fig. 6. Within each group, elements with the same precision are packed as integers, eliminating the need for additional padding, which saves space. Given that the bit-widths of integers is a power of 2, this is compatible with group size that is also a power of 2. For instance, even with the odd-bit such as 3-bit storage, integers can store these numbers without padding, as the commonly used group size is 128, a multiple of almost all definition of integer type. This ensures that elements within a group fully utilize the space provided by integers, without storing numbers of different precision within the same integer. $\hat{z}_{\text{int}}$ follow the original logic of AutoGPTQ but are packed with a uniform precision along the channel direction for ease of use. Other tensors, like scales, remain in the same floating-point format to ensure the correctness of dequantization calculations.

To indicate the precision of each group, we also introduce an additional array to store bit-widths of each group, where each number is represented as a 2-bit value aggregated into integers, marking the quantization precision of each group for accurate reconstruction. We use cumulative calculations to determine the starting index of each group, ensuring correctness despite changes in $\hat{w}_{\text{int}}$ height and starting indices caused by varying precision. Using the above methods to store the quantized weights, zeros, and additional bit arrays effectively reduces memory usage during model storage and loading, thereby lowering the resource overhead required for model deployment.

Once the weights are packed, we follow the modified AutoGPTQ logic for GPU inference. The GPU processes and dequantizes the weights group by group for computation. During GPU computation, a thread dequantizes a segment of continuous memory data in one column of $\hat{w}_{\text{int}}$ and performs vector dot product calculations with the input activation shared within the block, accumulating the results in the corresponding result matrix. When threads form a logical block, the block handles the computation and reduction of a continuous channel region. We complete the linear layer computation by iterating through all logical blocks. Leveraging AutoGPTQ's initial logic and CUDA Warp's 32-thread units, we ensure similar code structure and data access logic for threads within each warp when group size is 128. This method was primarily conducted to validate feasibility os SliM-LLM, demonstrating that the mixed precision quantization with integer packing does not cause additional computational overhead, indicating the efficiency and accuracy advantage of SliM-LLM. In summary, by dividing weight into several structured groups with mixed precision and employing a reasonable GPU utilization strategy, Slim-LLM balances performance and efficiency.

**Algorithm 1** Main Framework of SliM-LLM.

func SliM-LLM($\boldsymbol{w}, \boldsymbol{x}_F, \beta, \lambda, N$)
**Input:** $\boldsymbol{w} \in \mathbb{R}^{n \times m}$ - FP16 weight
$\qquad \boldsymbol{x}_F \in \mathbb{R}^{t \times m}$ - calibration data
$\qquad \beta$ - group size
$\qquad \lambda$ - hessian regularizer
$\qquad N$ - average bit-width
**Output:** $\hat{w}_q$ - quantized weight

1: $\boldsymbol{H} := \frac{1}{P} \sum_{k=1}^{P} \boldsymbol{x}_F^{[k]} \boldsymbol{x}_F^{[k]T}$ hessian matrix
2: $\boldsymbol{H}^{\text{in}} := \text{Cholesky}((\boldsymbol{H} + \lambda \boldsymbol{I})^{-1})$
3: $\hat{w}_q := 0^{n \times m}$

4: $\mathcal{G}\{\cdot\} := \text{SBA}(\boldsymbol{w}, \boldsymbol{x}_F, \boldsymbol{H}^{\text{in}}, \beta, N)$
5: **for** $b = 0, \beta, 2\beta, ...$ **do**
6: $\qquad \boldsymbol{w}^b := \boldsymbol{w}_{:,b:b+\beta}$
7: $\qquad g_b := \mathcal{G}[b]$
8: $\qquad \boldsymbol{w}_s^b, \boldsymbol{w}_{us}^b := \text{sal\_mask}(\boldsymbol{w}^b)$
9: $\qquad \hat{\boldsymbol{w}}_q^b := \text{SQC}(\boldsymbol{w}_s^b, \boldsymbol{w}_{us}^b, g_b)$
10: $\qquad$ *GPTQ-error compensation:*
11: $\qquad \boldsymbol{E} := (\boldsymbol{w}_{:,b:b+\beta} - \hat{\boldsymbol{w}}_q^b)/\boldsymbol{H}_{bb:b+\beta b+\beta}^{\text{in}}$
12: $\qquad \boldsymbol{w}_{:,b+\beta:} := \boldsymbol{w}_{:,b+\beta:} - \boldsymbol{E} \cdot \boldsymbol{H}_{b:b+\beta,b+\beta:}^{\text{in}}$
13: **end for**
14: **return** $\hat{w}_q$

---

**Algorithm 2** Detailed functions in SliM-LLM.

func SBA($\boldsymbol{w}, \boldsymbol{x}_F, \boldsymbol{H}^{\text{in}}, \beta, N$)

1: $\mathcal{G}\{\cdot\} := \{0\}$ // initialize group bit-width
2: $e := \inf$ // bit-widths searching error
3: $p^* := 0$ // number of ($N$-1)-bit and ($N$+1)-bit
4: $l := N - 1$ // lower bit-width
5: $h := N + 1$ // higher bit-width
6: $S\{\cdot\} := \text{average}(\frac{\boldsymbol{w}^2}{[\boldsymbol{H}^{\text{in}}]_{\text{diag}}^2})$
7: **for** $p = 1, 2, ..., [\frac{m}{2\beta}]$ **do**
8: $\qquad \hat{\boldsymbol{w}}_l^b := \text{fakequant}(\boldsymbol{w}_{b \in \text{top\_k\_min(p)}}^b, l, )$
9: $\qquad \hat{\boldsymbol{w}}_h^b := \text{fakequant}(\boldsymbol{w}_{b \in \text{top\_k\_max(p)}}^b, h, )$
10: $\qquad \hat{\boldsymbol{w}}_N^b := \text{fakequant}(\boldsymbol{w}_{b \in \text{others}}^b, N, )$
11: $\qquad \hat{\boldsymbol{w}}_q := \hat{\boldsymbol{w}}_l^b \cup \hat{\boldsymbol{w}}_l^b \cup \hat{\boldsymbol{w}}_h^b$
12: $\qquad$ **if** $\mathcal{D}_{kl}(\boldsymbol{x}\boldsymbol{w}^\top \| \boldsymbol{x}\hat{\boldsymbol{w}}_q^\top) < e$ **then**
13: $\qquad\qquad e := \mathcal{D}_{kl}(\boldsymbol{x}\boldsymbol{w}^\top \| \boldsymbol{x}\hat{\boldsymbol{w}}_q^\top)$
14: $\qquad\qquad p^* := p$
15: $\qquad$ **end if**
16: **end for**
17: $\mathcal{G}\{l\} := S\{\text{top\_k\_min}(p^*) = l\}$
18: $\mathcal{G}\{h\} := S\{\text{top\_k\_max}(p^*) = h\}$
19: $\mathcal{G}\{N\} := S\{\text{middle\_k}([\frac{m}{2}] - 2p^*) = N\}$
20: **return** $\mathcal{G}\{\cdot\}$

func SQC($\boldsymbol{w}_s^b, \boldsymbol{w}_{us}^b, g_b$)

1: $w_{\max} := \max(\boldsymbol{w}_s^b \cup \boldsymbol{w}_{us}^b)$
2: $w_{\min} := \min(\boldsymbol{w}_s^b \cup \boldsymbol{w}_{us}^b)$
3: $\lambda := 0.1$
4: $n := 50$
5: $e := \inf$ // scale searching error
6: $\Delta^* \in \mathbb{R}^{n \times 1}$ // per-channel scale
7: $z^* \in \mathbb{R}^{n \times 1}$ // per-channel zero point
8: **for** $\tau \in [1 - \lambda, 1 + \lambda]$ with $2n$ slices **do**
9: $\qquad \Delta := \tau(w_{\max} - w_{\min})/(2^{g_s} - 1)$
10: $\qquad z := -\lfloor (\tau w_{\min})/\Delta \rceil$
11: $\qquad \hat{\boldsymbol{w}}_s^b := \text{fakequant}(\boldsymbol{w}_s^b, g_b, \Delta, z)$
12: $\qquad \hat{\boldsymbol{w}}_{us}^b := \text{fakequant}(\boldsymbol{w}_{us}^b, g_b, \Delta, z)$
13: $\qquad \mathcal{L}_s := \|\boldsymbol{w}_s^b - \hat{\boldsymbol{w}}_s^b\|^2$
14: $\qquad \mathcal{L}_{us} := \|\boldsymbol{w}_{us}^b - \hat{\boldsymbol{w}}_{us}^b\|^2$
15: $\qquad$ **if** $\mathcal{L}_s + \mathcal{L}_{us} < e$ **then**
16: $\qquad\qquad e := \mathcal{L}_s + \mathcal{L}_{us}$
17: $\qquad\qquad z^* := z$
18: $\qquad\qquad \Delta^* := \Delta$
19: $\qquad$ **end if**
20: **end for**
21: $\hat{\boldsymbol{w}}_q^b := \text{fakequant}(\boldsymbol{w}^b, g_b, \Delta^*, z^*)$
22: **return** $\hat{\boldsymbol{w}}_q^b$

---

## C. Searching Details of Group-Wise Salience-Determined Bit Allocation

We optimize the mixed-precision configuration based on the output information entropy (KL-divergence), searching for the optimal compensation bit-widths ratio as shown in Eq. (4).

Initially, we rank each group by their average salience, a metric for quantization, and employ a double-pointer that moves simultaneously from both the beginning (lowest salience) and end (highest salience) of the sorted list. This ensures an equal number of groups at low and high bit-widths, effec-tively balancing the global average bit-widths compensation. We then calculate the relative entropy under the corresponding precision ratio and search for the optimal ratio. Fig 7 displays the search error curves related to the $2^{nd}$, $10^{th}$, and $15^{th}$ Transformer layers in the OPT1.3B model, showcasing the search curves for certain self-attention layers (Query, Key, Value, FC2).

Due to the limited range of the search, extreme scenarios involve either a half ($N - 1$)-bit and half ($N + 1$)-bit without $N$-bit or all groups being $N$-bit (uniform precision). Fig 7 demonstrates that lower quantization errors can be

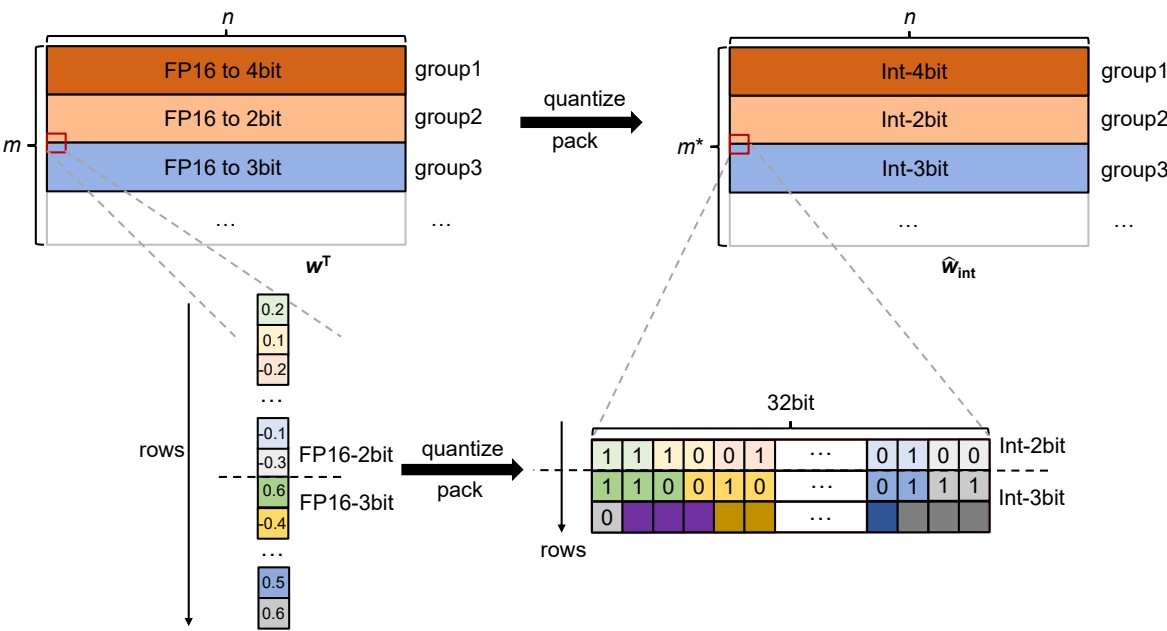

*Figure 6.* The memory layout shown in the figure is modified based on AutoGPTQ. The transposed original weights $\boldsymbol{w}^{\top} \in \mathbb{R}^{m \times n}$ are still divided into multiple groups along the row direction after quantization. The elements within each group are vertically packed into integers and then reassembled into $\hat{\boldsymbol{w}}_{\text{int}}$. The figure employs corresponding colors to indicate how each original number is mapped to a specific position within the packed integers after quantization, which finally generates $\hat{\boldsymbol{w}}_{\text{int}} \in \mathbb{R}^{m^{*} \times n}$, where $m^{*}$ is compressed from $m$ by packing several low-bit number. Similarly, $\hat{\boldsymbol{z}}_{\text{int}}$ is also packed into integers to save memory.

achieved under mixed-precision compared to quantization at the uniform bit-width. We also find that multiple low-error precision combinations are possible within a group of weights, allowing SBA to flexibly select the optimal ratio through its versatile search.

## D. Evluatiion Function of SBA

In Tab. 6, we employ various objective functions and compare their performance in SBA across different models. Compared to the commonly used Mean Squared Error (MSE) loss, Kullback-Leibler (KL) divergence ensures the distribution of critical activation positions within the model from an information entropy perspective, making it a superior choice for the bit-widths allocation strategy in SBA for the OPT and LLaMA models. When computing KL divergence in this context, we first transform the layer outputs into probability distributions using softmax.

## E. Extension Ablation on SQC

In this section, we visualize the effectiveness of SQC in mitigating the degradation of information in locally salient weights. We observed the absolute error of weights in a randomly selected channel of the quantized OPT-1.3B model. As shown in Fig. 8, the overall absolute error of the weights post-quantization with a standard quantizer was

0.0055, while with SQC it was reduced to 0.0039. This further demonstrates that the search parameter $\tau$, as applied in Eq. (5), effectively optimizes the quantizer parameters, thereby reducing quantization errors.

More importantly, SQC effectively perceives the information of locally salient weights, as indicated by the red regions in Fig. 8. Compared to the vanilla quantizer, SQC significantly reduces the error of salient weights. Specifically, the prominent weights at indices 375 in Fig. 8(a) show higher quantization errors, while in Fig. 8(b), this error is effectively reduced. This confirms SQC's ability to perceive locally salient weights, effectively preventing the degradation of critical information.

## F. Extension Ablation on Quantization Group-Size

To investigate the impact of different group sizes on the quantization effectiveness of SliM-LLM, we evaluated performance with 256 and 512 columns at a 3-bit level, observing that larger group sizes enhance GPU efficiency during inference. The findings suggest that increased group granularity does not substantially elevate perplexity across four models, indicating that SliM-LLM is robust and conducive to more efficient deployment methods. In contrast, at 2-bit, we assessed group sizes of 64 and 32 columns. With finer group granularity, the models displayed reduced perplex-

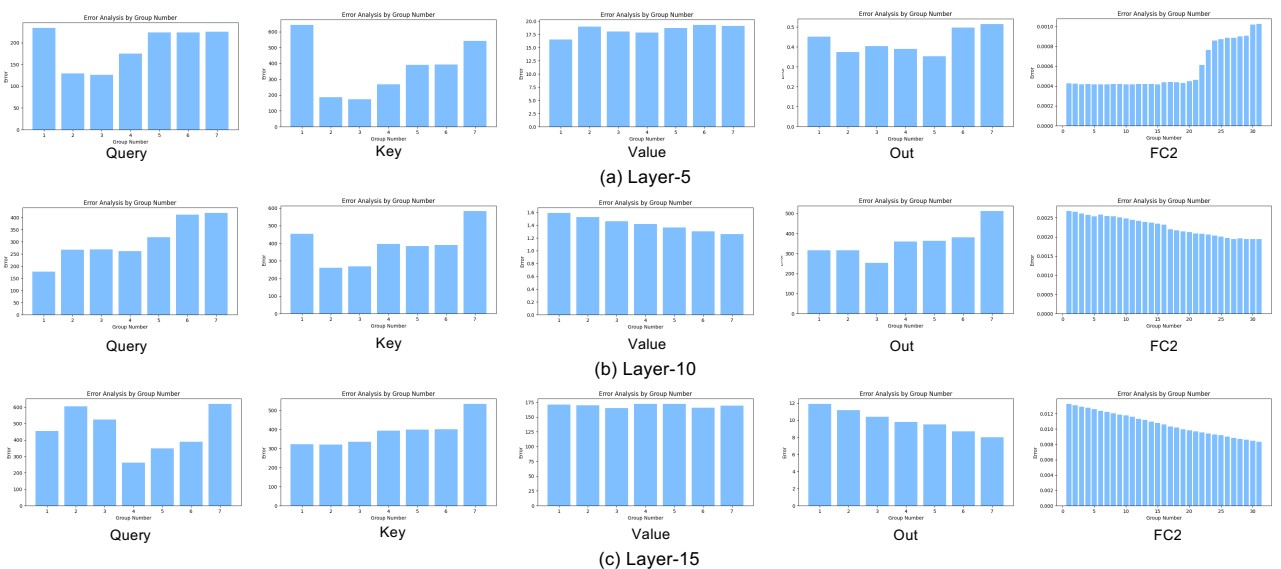

*Figure 7.* Error curves of SBA for select weights in the $5^{th}$, $10^{th}$, and $15^{th}$ layers of OPT-1.3B.

*Table 6.* Comparison of MSE and KL Divergence in SBA.

| Method | # W | OPT-1.3B | OPT-2.7B | OPT-6.7B | OPT-13B | LLaMA-7B | LLaMA2-7B |
|--------|-----|----------|----------|----------|---------|----------|-----------|
| MSE | 2-bit | 32.50 | 27.58 | 15.14 | 13.28 | 21.94 | 16.86 |
| **KL Divergence** | 2-bit | **30.71** | **13.26** | **11.27** | **10.12** | **14.58** | **16.01** |

*Table 7.* Ablation results on OPT-6.7B, LLaMA-7B, LLaMA-2-7B, LLaMA-3-8B with SliM-LLM under different group size (#g denotes the group size).

| Precision / PPL↓ | #g | OPT-6.7B | LLaMA-7B | LLaMA-2-7B | LLaMA-3-8B |
|------------------|-----|----------|----------|------------|------------|
| | 512 | 11.65 | 6.96 | 6.69 | 8.87 |
| 3-bit | 256 | 11.33 | 6.92 | 6.94 | 8.14 |
| | 128 | 11.27 | 6.40 | 6.24 | 7.62 |
| | 128 | 14.41 | 14.58 | 16.01 | 39.66 |
| 2-bit | 64 | 13.95 | 13.41 | 15.02 | 29.84 |
| | 32 | 12.47 | 11.91 | 11.95 | 16.93 |

ity. This is attributed to smaller groups providing more detailed data representation and utilizing additional quantization parameters, although they also raise computational and storage demands. A group size of 128 strikes a better balance between efficiency and quantization performance.

# G. Extension on Salience Channel Clustering

## G.1. Discussion of Theorem 1

*Theorem* 1. Given the input calibration activation $\boldsymbol{x} \in \mathbb{R}^{t \times m}$ with an outlier channel $\boldsymbol{x}^*_{:,p} \gg \boldsymbol{x}_{:,j}, \forall j \in [0, m], j \neq p$ at the position of channel-$p$. The trace elements of $\boldsymbol{H} = \boldsymbol{x}^\top \boldsymbol{x}$ will show great outlier value at $(p, p)$, where $\boldsymbol{H}_{p,p} \gg \boldsymbol{H}_{j,j}, \forall j \in [0, m], j \neq p$, as $\boldsymbol{H}_{p,p}$ is produced by $[\boldsymbol{x}^{*\top}_{:,p} \boldsymbol{x}^*_{:,p}] = \sum_{i=0}^{t} x_{i,p}^{*2}$, which further leads to the pa-

rameter salience larger at the $p^{th}$ channel of weight, where $\delta_{:,p} > \delta_{:,k}, \delta_{:,k} = \frac{w_{:,k}^2}{[\boldsymbol{H}^{-1}]_{k,k}^2}, \forall k \in [0, t], k \neq p$.

*Proof.* Given $\boldsymbol{x} \in \mathbb{R}^{t \times m}$ with outlier channel $\boldsymbol{x}^*_{:,p}$, $p \in [0, m]$, and other elements with small magnitude $x_{i,j}$, where $x^*_{q,p} \gg x_{i,j}$ and $i, j \neq q, p$. We can get the Hessian matrix with Levenberg-Marquardt ([Marquardt], [1963]) approximation in Eq. ([3]):

$$\boldsymbol{H} = \begin{pmatrix} x_{11}^2 + .. & \cdots & \cdots & \cdots \\ \vdots & \ddots & \cdots & \vdots \\ \vdots & \vdots & \boldsymbol{x}^*_{\boldsymbol{p},\boldsymbol{p}}{}^2 + .. & \vdots \\ \cdots & \cdots & \cdots & \ddots \end{pmatrix} \quad (6)$$

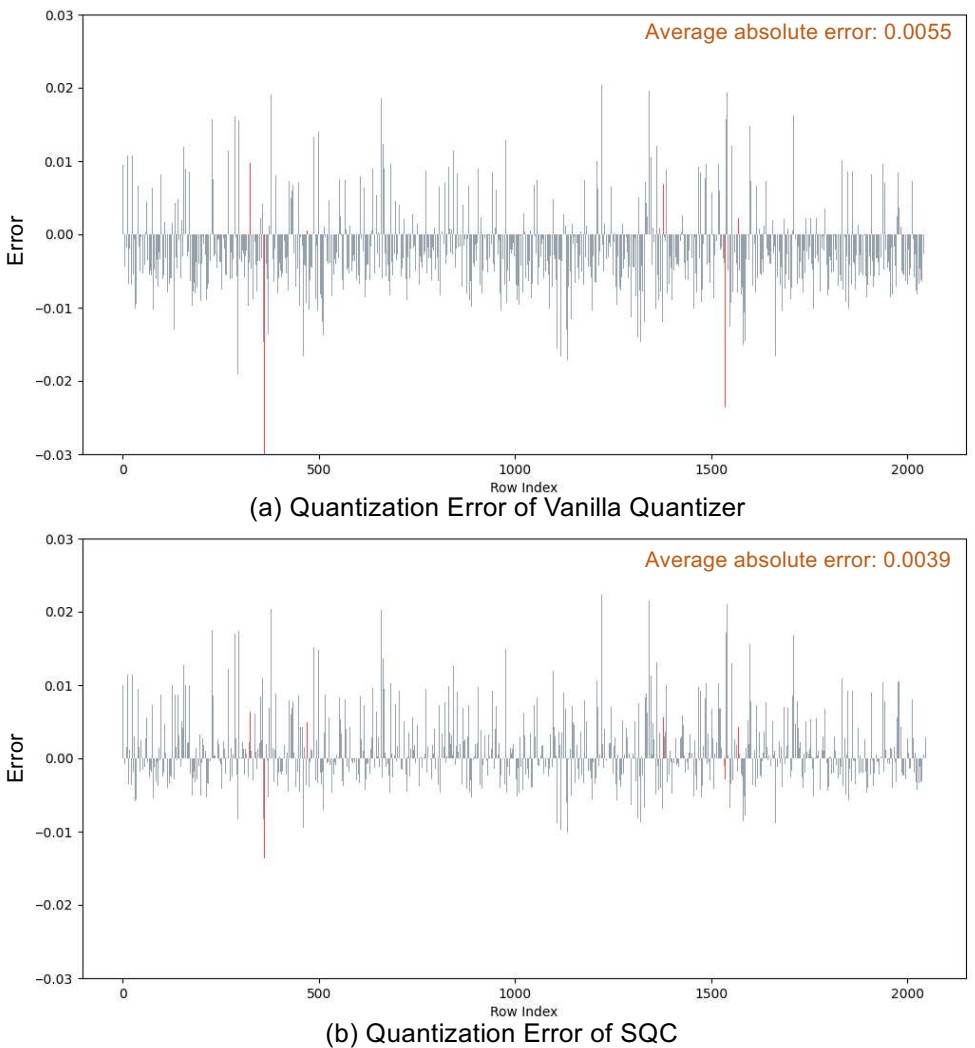

*Figure 8.* Absolute channel error of the weight of the OPT-1.3B model. The red line represents the quantization error for the locally salient weights, and the lightmauve represents other weights. (a) Vanilla quantizer error on the $794^{th}$ channel of OPT-1.3B. (b) SQC error on the $794^{th}$ channel of OPT-1.3B

where $[\boldsymbol{x}^{*\top}_{:,\boldsymbol{p}} \boldsymbol{x}^{*}_{:,\boldsymbol{p}}]$ will appears at position $\boldsymbol{H}_{p,p}$. And following SparseGPT (Frantar & Alistarh, 2023), the inverse matrix of $\boldsymbol{H}$ can be formulated as:

$$\delta_{i,j} = \frac{w^2_{i,j}}{[\text{diag}((\boldsymbol{x}^\top \boldsymbol{x} + \lambda \boldsymbol{I})^{-1})]^2} \quad (7)$$

where $(\boldsymbol{x}^\top \boldsymbol{x} + \lambda \boldsymbol{I})^{-1}$ is the new representation of Hessian matrix $\boldsymbol{H}$ for the layer-wise reconstruction problem, and $\lambda$ is the dampening factor for the Hessian to prevent the collapse of the inverse computation. Additionally, in accordance with the configuration in LLMs (Frantar & Alistarh, 2023; Frantar et al., 2022; Sun et al., 2023), the value of $\lambda$ set is extremely small ($\lambda \leq \mathrm{e}^{-1}$), while the values located at the diagonal of Hessian are large. Therefore, only considering the influence of diagonal elements (Sun et al., 2023),

we can further approximate salience as:

$$\delta_{i,j} = \frac{w^2_{i,j}}{[\text{diag}((\boldsymbol{x}^\top \boldsymbol{x} + \lambda \boldsymbol{I})^{-1})]^2} \approx \\ \frac{w^2_{i,j}}{[(\text{diag}(\boldsymbol{x}^\top \boldsymbol{x}))^{-1}]^2} = (w_{i,j} \cdot ||\boldsymbol{x}_j||^2_2)^2 \quad (8)$$

Here the diagonal of $\boldsymbol{x}^\top \boldsymbol{x}$ is $\text{diag}(||\boldsymbol{x}_j||^2_2)$, and $||\boldsymbol{x}_j||_2$ evaluates the $\ell_2$ norm of $j^{th}$ channel across different tokens. Consequently, it can be summarized that when there is an outlier channel-$p$, the value of $||\boldsymbol{x}_p||_2$ is primarily influenced by $[\boldsymbol{x}^{*\top}_{:,\boldsymbol{p}} \boldsymbol{x}^{*}_{:,\boldsymbol{p}}]$. Additionally, since the activation values are relatively large and the differences in weight values are comparatively small, the $p^{th}$ channel of weights will also exhibit salience. $\qquad\square$

## G.2. Distribution of salience, activation and weight magnitude

Fig. 9 illustrates the distribution of salience among certain weights in LLMs. This section provides additional examples to demonstrate how the distribution of weights and input activation characteristics influence the salience of parameters in LLMs. The figure captures seven linear projections in the multi-head self-attention (MHA) and feed-forward block (FFB) layers of the $2^{nd}$ and $10^{th}$ Transformer modules in the LLaMA-7B model.

In line with previous findings (Nrusimha et al., 2024; Xiao et al., 2023a), activations demonstrate particularly marked outlier phenomena on anomalous tokens and channels, with extremes differing by more than two orders of magnitude. Notably, distinct anomalous channels are present in the MHA's Query, Key, and Value layers, where outliers vary significantly across different tokens. This pattern is consistent in the FFB layers. We observe that disparities in weight magnitudes are less pronounced than those in activation, thus exerting a reduced impact on outlier channels. Moreover, weights distribute structurally along rows or columns (Dettmers et al., 2023; Huang et al., 2024a), affecting the overall distribution of salience from a row-wise perspective (Fig. 9). However, the most prominent salience is predominantly driven by activation across channels (column-wise).

## G.3. Hessian Diagonal Clustering

Sec. 3.2.1 demonstrates that outlier tokens in input activations result in significant values at the corresponding positions along the diagonal of the weight Hessian matrix. Additionally, due to the token sink phenomenon (Xiao et al., 2023b; Nrusimha et al., 2024), areas around significantly activated key tokens exhibit increased salience, creating clusters of salient regions along the Hessian matrix diagonal. To further elucidate this phenomenon, Fig. 10 shows the values along the diagonal of the Hessian matrix for selected weights in the $2^{nd}$ and $10^{th}$ layers of the LLaMA-7B model. Within this diagonal, certain positions display pronounced values (indicated in red), whereas others are relatively moderate. In the attention aggregation layer of the $10^{th}$ layer, the token sink phenomenon results in a pronounced convergence of significant values along the Hessian matrix diagonal, with deep red areas indicating regional clustering. These findings reinforce the influence of input activations on the diagonal of the Hessian matrix, subsequently leading to a clustering phenomenon in the salience distribution of weights across channels.

## H. More Comparisons

In this section, we provide supplementary experiments for SliM-LLM. Tab. 8 displays the comparative results of SliM-LLM and SliM-LLM$^+$ with other methods on the OPT series models. Tab. 9 shows the performance of SliM-LLM when quantizing the LLaMA family models on the C4 dataset, while Tab. 10 also compares the results of SliM-LLM$^+$ on the C4 dataset. In Tab. 11, we compared the quantization results of GPTQ, AWQ, and SliM-LLM at 2-bit on the Gemma2 and Mixtral models, demonstrating the greater stability of SliM-LLM across a wider range of model structures. Additionally, in Tab. 12, we supplemented the 4-bit results of different quantization methods in the LLaMA series models, showing that SliM-LLM and SliM-LLM$^+$ exhibit the smallest quantization errors at practical 4-bit levels. To provide a comprehensive evaluation across a broader set of benchmarks, we further compared the quantization results on MMLU and MathQA in Tab. 13.

## I. Real Dialog Examples

In this section, we show some dialogue examples of LLaMA-2-13B and Vicuna-13B with SliM-LLM-2bit and GPTQ-2bit in Fig. 11.

## J. Model Experiments on Efficiency

Tab. 14 presents further efficiency comparisons between AutoGPTQ and SliM-LLM. The results shows that even under the larger LLMs with 70B parameters, SliM-LLM can also show the better accuracy with comparable inference efficiency.

## K. Quantization Performance on Vision Language Models

To further showcase the potential application capabilities of SliM-LLM, in Tab. 15, we deploy SliM-LLM on LLaVA-Next-8B (Liu et al., 2024) evaluated on 4 benchmarks. The results show that GPTQ, AWQ and SliM-LLM show comparable performance under the 3-bit context. However, when the bit-width is setting as 2, GPTQ and AWQ failed to generate the reasonable answer in each benchmark and get "N" results. SliM-LLM successfully generate the reasonable output and the accuracy is closed to 3-bit model, which presents the superior usability of SliM-LLM to wider environments.

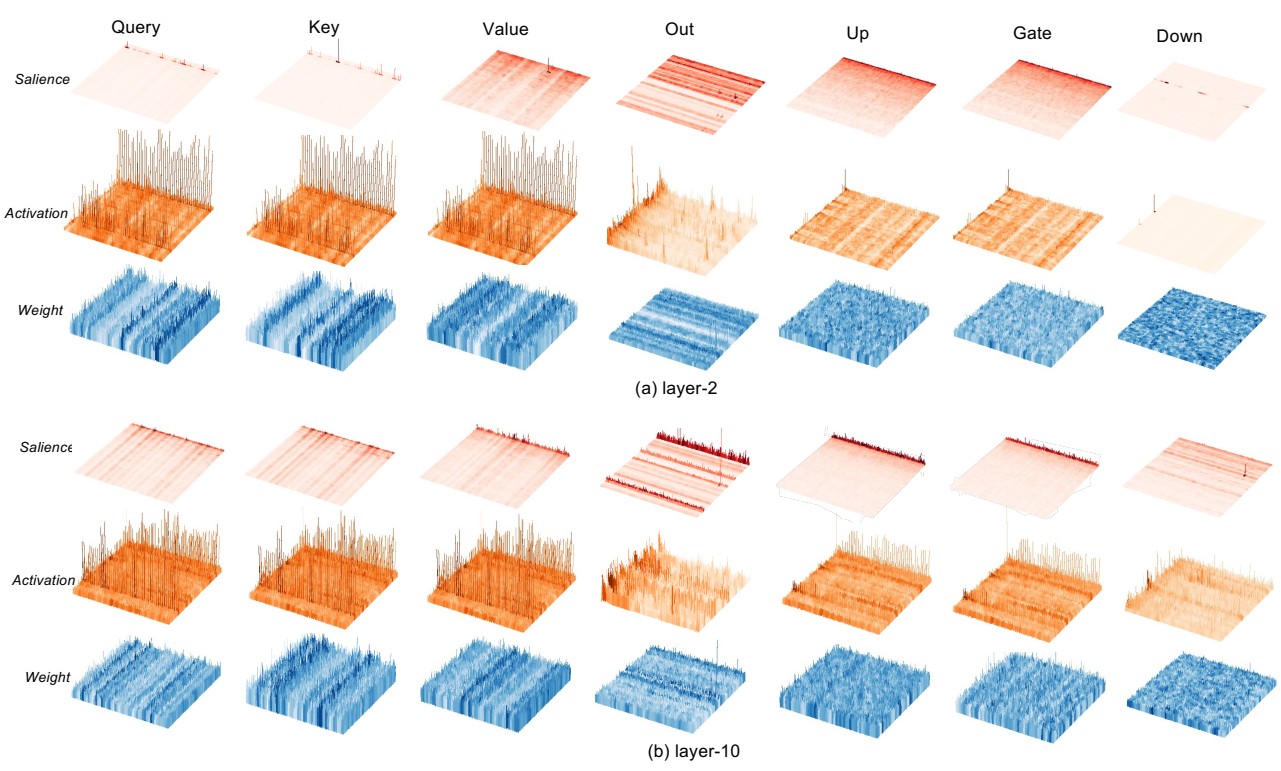

Query  Key  Value  Out  Up  Gate  Down

*Salience*

*Activation*

*Weight*

(a) layer-2

*Salience*

*Activation*

*Weight*

(b) layer-10

*Figure 9.* Salience, activation and weight distribution in the $2^{nd}$ and $10^{th}$ layers of LLaMA-7B

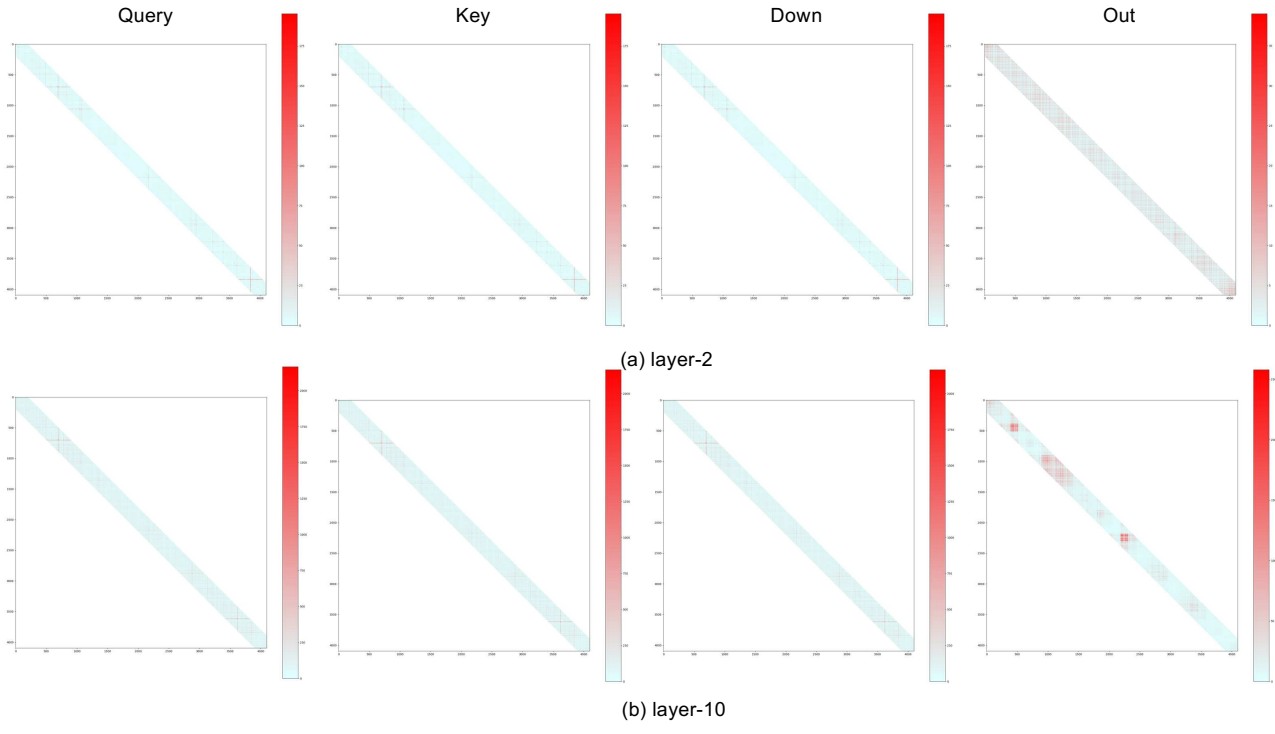

Query  Key  Down  Out

(a) layer-2

(b) layer-10

*Figure 10.* Hessian diagonal magnitude in attention layers of $2^{nd}$ and $10^{th}$ layers of LLaMA-7B

*Table 8.* Quantization results of OPT Models on WikiText2 (group size is 128).

| #W PPL↓ | Method | 1.3B | 2.7B | 6.7B | 13B | 30B | 66B |
|---|---|---|---|---|---|---|---|
| 16-bit | - | 14.63 | 12.47 | 10.86 | 10.12 | 9.56 | 9.34 |
| 3-bit | RTN | 1.2e2 | 3.0e2 | 23.54 | 46.03 | 18.80 | 1.4e6 |
| | GPTQ | 16.47 | 13.69 | 11.65 | 10.35 | 9.73 | 10.96 |
| | AWQ | 16.32 | 13.58 | 11.41 | 10.68 | 9.85 | 9.60 |
| | QuIP | 16.21 | 13.79 | 11.51 | 10.50 | 9.75 | 9.59 |
| | **SliM-LLM** | **15.91** | **13.26** | **11.27** | **10.26** | **9.70** | **9.48** |
| | OmniQuant | 15.72 | 13.18 | 11.27 | 10.47 | 9.79 | 9.53 |
| | AffineQuant | 15.61 | 12.98 | 11.18 | 10.51 | 9.81 | - |
| | **SliM-LLM$^+$** | **15.58** | **12.84** | **11.18** | **10.44** | **9.67** | **9.51** |
| 2-bit | RTN | 1.3e4 | 5.7e4 | 7.8e3 | 7.6e4 | 1.3e4 | 3.6e5 |
| | GPTQ | 1.1e2 | 61.59 | 20.18 | 21.36 | 12.71 | 82.10 |
| | AWQ | 47.97 | 28.50 | 16.20 | 14.32 | 12.31 | 14.54 |
| | QuIP | 41.64 | 28.98 | 18.57 | 16.02 | 11.48 | 10.76 |
| | PB-LLM | 45.92 | 39.71 | 20.37 | 19.11 | 17.01 | 16.36 |
| | **SliM-LLM** | **30.71** | **24.08** | **14.41** | **13.68** | **11.34** | **10.94** |
| | OmniQuant | 23.95 | 18.13 | 14.43 | 12.94 | 11.39 | 30.84 |
| | **SliM-LLM$^+$** | 24.57 | **17.98** | **14.22** | **12.16** | **11.27** | 14.98 |

*Table 9.* Quantization results of LLaMA Family with statistic quantizer on C4 (group size is 128).

| #W PPL↓ | Method | 1-7B | 1-13B | 1-30B | 1-65B | 2-7B | 2-13B | 2-70B | 3-8B | 3-70B |
|---|---|---|---|---|---|---|---|---|---|---|
| 16-bit | - | 7.08 | 6.61 | 5.98 | 5.62 | 6.97 | 6.46 | 5.52 | 9.22 | 6.85 |
| 3-bit | APTQ | 6.24 | - | - | - | - | - | - | - | - |
| | RTN | 8.62 | 7.49 | 6.58 | 6.10 | 8.40 | 7.18 | 6.02 | 1.1e2 | 22.39 |
| | AWQ | 7.92 | 7.07 | 6.37 | 5.94 | 7.84 | 6.94 | - | 11.62 | 8.03 |
| | GPTQ | 7.85 | 7.10 | 6.47 | 6.00 | 7.89 | 7.00 | 5.85 | 13.67 | 10.52 |
| | **SliM-LLM** | **6.14** | **6.05** | **6.33** | **5.94** | **7.74** | **5.26** | **5.09** | **13.10** | **8.64** |
| 2-bit | RTN | 1.0e3 | 4.5e2 | 99.45 | 17.15 | 4.9e3 | 1.4e2 | 42.13 | 2.5e4 | 4.6e5 |
| | AWQ | 1.9e5 | 2.3e5 | 2.4e5 | 7.5e4 | 1.7e5 | 9.4e4 | - | 2.1e6 | 1.4e6 |
| | GPTQ | 34.63 | 15.29 | 11.93 | 11.99 | 33.70 | 20.97 | NAN | 4.1e4 | 21.82 |
| | QuIP | 33.74 | 21.94 | 10.95 | 13.99 | 31.94 | 16.16 | 8.17 | 1.3e2 | 22.24 |
| | PB-LLM | 49.73 | 26.93 | 17.93 | 11.85 | 29.84 | 19.82 | 8.95 | 79.21 | 33.91 |
| | **SliM-LLM** | **32.91** | **13.85** | **11.27** | **10.95** | **16.00** | **9.41** | **7.01** | **1.1e2** | **15.92** |

*Table 10.* Quantization results of LLaMA-1 and LLaMA-2 models with learnable quantizer on C4.

| #W PPL↓ | Method | 1-7B | 1-13B | 1-30B | 1-65B | 2-7B | 2-13B | 2-70B |
|---|---|---|---|---|---|---|---|---|
| 16-bit | - | 7.08 | 6.61 | 5.98 | 5.62 | 6.97 | 6.46 | 5.52 |
| 3-bit | OmniQuant | 7.75 | 7.05 | 6.37 | 5.93 | 7.75 | 6.98 | 5.85 |
| | AffineQuant | 7.75 | 7.04 | 6.40 | - | 7.83 | 6.99 | - |
| | **SliM-LLM$^+$** | **7.75** | **6.91** | **6.36** | 5.96 | **7.71** | **6.90** | **5.85** |
| 2-bit | OmniQuant | 12.97 | 10.36 | 9.36 | 8.00 | 15.02 | 11.05 | 8.52 |
| | AffineQuant | 14.92 | 12.64 | 9.66 | - | 16.02 | 10.98 | - |
| | **SliM-LLM$^+$** | 14.99 | **10.22** | **9.33** | **7.52** | 18.18 | **10.24** | **8.40** |

*Table 11.* PPL Comparison on Gemma2 and Mixtral.

| Model/Evaluation | Method | PPL (wikitext2) |
|---|---|---|
| Gemma2-9B | GPTQ 2-bit | 186.77 |
| | AWQ 2-bit | 217.83 |
| | **SliM-LLM 2bit** | **26.30** |
| Mixtral 8x7B | GPTQ 2-bit | 16.38 |
| | AWQ 2-bit | 3.2e5 |
| | **SliM-LLM 2bit** | **7.44** |

*Table 12.* The PPL results of our proposed method and other methods under 4bit quantization.

| Method | LLaMA-7B | LLaMA-13B | LLaMA2-7B | LLaMA2-13B | LLaMA3-8B |
|---|---|---|---|---|---|
| FP16 | 5.68 | 5.09 | 5.47 | 4.88 | 5.75 |
| AWQ | 5.81 | 5.30 | 5.62 | 4.97 | 6.63 |
| GPTQ | 5.85 | 5.20 | 5.61 | 4.98 | 6.50 |
| **SliM-LLM** | **5.83** | **5.16** | **5.59** | **4.95** | **6.42** |
| Omniquant | 5.77 | - | 5.58 | - | - |
| **SliM-LLM$^+$** | **5.75** | - | **5.57** | - | - |

*Table 13.* The results(%) on MMLU and MathQA for multiple quantized LLaMA models.

| Model | Method | Humanities | Social Sciences | STEM | Other | MMLU | MathQA |
|---|---|---|---|---|---|---|---|
| LLaMA-7B | GPTQ 2-bit | 24.87 | 21.84 | 21.79 | 24.01 | 23.32 | 21.11 |
| | AWQ 2-bit | 24.21 | 21.71 | 21.25 | 23.98 | 22.95 | 22.21 |
| | **SliM-LLM 2bit** | **24.94** | **23.60** | **23.40** | **25.50** | **25.10** | **23.74** |
| LLaMA-13B | GPTQ 2-bit | 24.23 | 23.20 | 22.99 | 24.78 | 23.85 | 21.68 |
| | AWQ 2-bit | 24.17 | 31.07 | 28.61 | 25.14 | 26.89 | 21.98 |
| | **SliM-LLM 2bit** | **25.12** | **31.74** | **29.19** | **26.17** | **27.05** | **23.17** |
| LLaMA2-7B | GPTQ 2-bit | 25.02 | 22.13 | 22.61 | 23.17 | 23.44 | 21.07 |
| | AWQ 2-bit | 25.12 | 22.79 | 24.26 | 24.01 | 24.51 | 19.06 |
| | **SliM-LLM 2bit** | **26.60** | **23.23** | **25.70** | **25.70** | **25.81** | **22.55** |
| LLaMA2-13B | GPTQ 2-bit | 23.91 | 27.17 | 26.10 | 25.78 | 25.53 | 20.87 |
| | AWQ 2-bit | 24.17 | 31.07 | 28.61 | 25.14 | 26.89 | 19.53 |
| | **SliM-LLM 2bit** | **26.27** | **32.20** | **29.98** | **26.46** | **27.34** | **23.48** |

*Table 14.* LLaMA-2-70B results of GPTQ and Slim-LLM on GPU. Group size is set to 128.

| #W | LLaMA-2-70B | WM | RM | PPL↓ | Token/s |
|---|---|---|---|---|---|
| 3-bit | GPTQ | 28.0G | 34.9G | 3.85 | 6.5 |
| | **SliM-LLM** | 28.0G | 35.2G | **3.67** | 6.2 |
| 2-bit | GPTQ | 16.4G | 23.3G | 8.78 | 9.7 |
| | **SliM-LLM** | 23.5G | 4.4G | **6.28** | 8.4 |

*Table 15.* Quantization results on Vision Language Models (VLMs).

| | #W | #G | AI2D | ChartQA | DocVQA | MMBench |
|---|---|---|---|---|---|---|
| GPTQ | 3 | 128 | 66.2 | 65.1 | 75.6 | 67.4 |
| | 2 | 128 | N | N | N | N |
| AWQ | 3 | 128 | 67.7 | 65.4 | 74.4 | 68.0 |
| | 2 | 128 | N | N | N | N |
| **SliM-LLM** | 3 | 128 | **68.2** | **67.5** | **74.8** | **68.9** |
| | 2 | 128 | **57.2** | **49.3** | **60.6** | **60.9** |

## LLaMA-2-13B

|  |  |
|---|---|
| Prompt: | The eight planets in Solar System are Mercury, ____ |
| **GPTQ-2bit**: | , 10.2. The eight pill o free games of p. 62.2. The way we do this. |
| **Slim-LLM-2bit**: | Venus, Earth, Mars, Jupiter, Saturn, Uranus and Neptune. |
| Prompt: | The tallest and most famous tower in France is ____ |
| **GPTQ-2bit**: | is 350 and most recently. |
| **Slim-LLM-2bit**: | 320 meters high. |
| Prompt: | The 2012 Summer Olympic was held in ___ |
| **GPTQ-2bit**: | The 2012 Summer Olympic was held in 1016 (a.185) and the 775,000 km from each. |
| **Slim-LLM-2bit**: | 2012 in London, the United Kingdom. |

## Vicuna-13B

|  |  |
|---|---|
| Prompt: | What are the capitals of Brazil, Mexico and France? |
| **GPTQ-2bit**: | What is the RER score? |
| **Slim-LLM-2bit**: | The capital of Brazil is Brasília, the capital of Mexico is Mexico City, and the capital of France is Paris. |
| Prompt: | What are the ancient civilizations in Europe? |
| **GPTQ-2bit**: | I am trying to find a book to nook over the life of to 300,000 or so. The ancient civilization is the one in which history books be pop and I have a be, oh, do you know the one the Mayan, the one of the total for Dake. |
| **Slim-LLM-2bit**: | The ancient civilizations in Europe include the Roman Empire, the Byzantine Empire, and the Ottoman Empire. |
| Prompt: | What are the logical circuits? |
| **GPTQ-2bit**: | What is the order of the logical cumulated? What is the best strategy to do in a single? What is the most popular to be eaten? What are the most sought? |
| **Slim-LLM-2bit**: | The logical circuits are the basic components for building the logical functions, which are required for digital computing. These circuits use logic structures and devices that allow the use of Boolean and multiple logic to implement the functional requirements as defined by a problem solving method of computer. In digital computing, circuits perform the logical tasks required for solving problems. |

*Figure 11.* Some examples of conversations. LLaMA-2-13B and Vicuna-13B are chosen to show the case of language supplementary and Q&A ability. And GPTQ-2bit is selected as the comparison. We color the text to show the reasonable or inappropriate responses.

