# OpenReview forum: "SliM-LLM: Salience-Driven Mixed-Precision Quantization for Large Language Models"
_ICML.cc/2025/Conference — ICML 2025 poster_

### Official Review · Reviewer_H7Hr · 2025-02-27

**Overall Recommendation:** 3

**Summary:**

This work proposes a mixed-precision quantization approach with coarse-level and fine-level partitioning via proposed Salience-Determined Bit Allocation and Salience-Weighted Quantizer Calibration, respectively. The former leverages the double-pointer search algorithm, optimizing KL-divergence between the original model and with the given weight quantized and the line search to determine local-level outliers. The proposed approach is compatible with different quantizers. Slim-LLM is evaluated on top of GPTQ and learnable quantizer Slim-LLM+ on Llama-1,2,3 model family.

**Claims And Evidence:**

The key aspect of the proposed method is the uneven bit width allocation according to weight saliency. While the introduced idea in the presented form is novel, the claim that it is largely overlooked in prior literature is inaccurate, as there are numerous methods that account for outliers in prior work via storing them in sparse format [1, 2] or orthogonal transformation [3, 4]. I would suggest restating the claim that this work proposes a new solution that accounts for uneven weight sensitivity.

The proposed saliency measure is identical to the one adopted in [1] to determine outliers and for sensitivity analysis (see Equation 2 and Figure 2). However, no reference is provided.

The presented empirical results show pretty reasonable performance for 2-bit quantization as compared to other uniform quantization methods.

---

**References**

[1] Dettmers, Tim, et al. "Spqr: A sparse-quantized representation for near-lossless llm weight compression." arXiv preprint arXiv:2306.03078 (2023).

[2] Kim, Sehoon, et al. "SqueezeLLM: Dense-and-Sparse Quantization." International Conference on Machine Learning. PMLR, 2024.

[3] Chee, Jerry, et al. "Quip: 2-bit quantization of large language models with guarantees." Advances in Neural Information Processing Systems 36 (2023): 4396-4429.

[4] Liu, Zechun, et al. "Spinquant: Llm quantization with learned rotations." arXiv preprint arXiv:2405.16406 (2024).

**Essential References Not Discussed:**

DB-LLM is mentioned in the related work, but not compared with.

**Experimental Designs Or Analyses:**

Overall, experimental protocol and choice of baselines is sensible.
However, the comparison with DB-LLM [1], leveraging mixed precision, a natural competitor to the proposed approach, is absent. I believe it should be added for completeness of evaluation.

---

**References**

[1] Chen, Hong, et al. "DB-LLM: Accurate Dual-Binarization for Efficient LLMs." Findings of the Association for Computational Linguistics ACL 2024. 2024.

**Methods And Evaluation Criteria:**

The evaluation protocol adopted is standard for research on LLM compression.

**Other Comments Or Suggestions:**

-

**Other Strengths And Weaknesses:**

While the approach yields quite good performance at 2 bit compression, the performance still lags behind vector quantization methods [1, 2], which achieve same or better speed-ups.

---

**References**

[1] Tseng, Albert, et al. "QuIP $\# $: Even Better LLM Quantization with Hadamard Incoherence and Lattice Codebooks." Forty-first International Conference on Machine Learning.

[2] Egiazarian, Vage, et al. "Extreme Compression of Large Language Models via Additive Quantization." Forty-first International Conference on Machine Learning.

**Questions For Authors:**

* Which dataset is used for the double-pointer search - is it the same calibration set used for SilM-LLM+ (i.e 128 samples from Wikitext-2)?

* How much does it take to produce a quantized model with SilM-LLM?

**Relation To Broader Scientific Literature:**

This work proposes a new way to account for uneven importance of weights via mixed precision with global and local criteria.

**Theoretical Claims:**

This work provides primarily a practical contribution. The theoretical motivation of the proposed approach is sound.

---

> ### Author Rebuttal · Authors · 2025-04-01
>
> Dear Reviewer H7Hr,
>
> Thank you for your feedback. We will address your questions and recommendations one by one.
>
> > Q1: The key aspect of the proposed method ... for uneven weight sensitivity. The proposed saliency measure ... However, no reference is provided.
>
> A: We would like to clarify that in lines 45–46 of the paper, we emphasize that weights "exhibit a structured distribution" a phenomenon that has not been overlooked in previous works. Prior studies mainly focused on compressing weights based on the sparse distribution of salience but did not propose specific strategies to handle structured distribution. This insight motivated us to develop an inference-friendly mixed-precision quantization strategy. Thank you for your suggestion—we will highlight the characteristics of structured distribution more clearly.
>
> Regarding SPQR, we explicitly reference SPQR (Line 199) in Definition 3.2 and also cite other works that adopt the same salience definition[1] [2] [3]. We would like to reiterate that the purpose of our work is not to emphasize the formulation of salience itself, but rather to identify its clustering characteristics within LLM weight matrices. SliM-LLM aims to illustrate, through Definition 3.2, the numerical impact of weight magnitude and activation distribution on the salience of LLM weight matrices and provides in-depth theoretical analysis in Theorem 1 and Appendix G, which further confirms the cause of the observed structured clustering of salience in SliM-LLM.
>
> [1]SparseGPT: Massive Language Models Can Be Accurately Pruned In One-shot. ICML, 2023.
>
> [2]PB-LLM: Partially Binarized Large Language Models. ACL 2024.
>
> [3]LLM-MQ: Mixed-Precision Quantization for Efficient LLM deployment. NIPS, 2024.
>
> > Q2: Overall, experimental protocol and choice of baselines ... be added for completeness of evaluation.
>
> A: As summarized in Figure 1, DB-LLM is a QAT-based method, which requires a significant amount of data and extended distillation tuning to complete the quantization process. In other words, it relies on standard backpropagation techniques to adjust the weights. In contrast, SliM-LLM is a totally PTQ method. More importantly, the key advantage of SliM-LLM is its ability to deploy group-wise quantization kernels directly on the GPU, and its seamless integration with AutoGPTQ. DB-LLM is an effective weight-compression method, focusing on memory compression, and is not yet capable of achieving efficient inference and real acceleration, which is the gap we aim to address.
>
> > Q3: While the approach yields quite good performance ...... [1, 2], which achieve same or better speed-ups.
>
> A: QuIP# and AQLM are both highly effective 2-bit LLM codebooks-based or vector-based quantization methods that can be improved by additional fine-tuning to restore performance (referenced in our paper). They rely on backpropagation to adjust pre-trained weights to fit the quantization scenario. In contrast, as we emphasize, SliM-LLM is a fully PTQ-based quantization method, with no training on the original weights. For a fair comparison of quantization strategies, we only compare with state-of-the-art PTQ methods, and we have achieved strong low-bit quantization performance within the PTQ framework.
>
> |#W PPL↓|Method|1-7B|1-13B|2-7B|2-13B|
> |-------|---------|----|-----|-----|-----|
> |2-bit|QuIP#|9.95|7.18|12.30|7.60|
> |2-bit|SliM-LLM+|9.68|7.18|10.87|7.59|
>
> Following your suggestion, we compared the non-trained QuIP# with our SliM-LLM+ at 2-bit. The results above show that even without additional weight tuning, SliM-LLM+ still achieves superior performance.
>
> Regarding the speed-ups you mentioned, we conducted a thorough investigation and reproduction of results. We found that codebook-based and vector-based quantization methods, while effective in compressing weight memory, require complex lookup and decoding operations during real deployment, resulting in inference times that are approximately three times longer than that of fp16 LLMs (https://github.com/Cornell-RelaxML/quip-sharp/issues/63). In contrast, the structured mixed-precision quantization strategy employed by SliM-LLM allows for efficient deployment in AutoGPTQ through a group-wise approach, achieving significant speed-ups.
>
> > Q4: Which dataset is used for the double-pointer search - is it the same calibration set used for SliM-LLM+ (i.e 128 samples from Wikitext-2)?
>
> A: Yes, as demonstrated in Section 4, we used the same calibration data setting (128 samples from WikiText-2), and during quantization, we selected samples randomly. The calibration selection method for SliM-LLM+ is identical to that of OmniQuant, both using random sampling of 128 data points from WikiText-2.
>
> > Q5: How much does it take to produce a quantized model with SliM-LLM?
>
> A: When applying SBA and SQC for PTQ mixed-precision quantization under single-GPU (RTX 4090) edge conditions, the quantization process for a 7B model takes approximately 25 minutes.

---

> > ### Comment · Reviewer_H7Hr · 2025-04-01
> >
> > After reading the rebuttal addressed to me and other reviewers, I decided to raise my score.
> >
> > While I still believe that a more accurate comparison with vector quantization methods—in terms of performance and speed-up—is necessary for one to fully appreciate the method’s efficacy and practicality, this is overall a decent work with the potential for practical deployment.

---

> > > ### Author Response · Authors · 2025-04-04
> > >
> > > Dear Reviewer H7Hr,
> > >
> > > We sincerely thank you for your thoughtful feedback and for raising your score. We truly appreciate your acknowledgment of the potential for practical deployment of our work. Your constructive critique has been invaluable in helping us refine our paper, and we are pleased to have addressed your concerns.
> > >
> > > We fully agree on the importance of a more accurate comparison with PTQ vector-quantization methods to thoroughly evaluate the efficacy and practicality of our approach. In response to your valuable suggestion, we will include a detailed comparison of SliM-LLM+ with the non-training QuIP# method (in terms of speed and accuracy) in our revised version.
> > >
> > > Thank you again for your engagement and support !

---

### Official Review · Reviewer_iFeq · 2025-03-12

**Overall Recommendation:** 3

**Summary:**

The paper introduces SliM-LLM, a novel PTQ framework for LLMs. The proposed method leverages the authors’ observation that important weights follow a structured distribution to preserve the model performance at extremely low-bit precision. Their two key contributions are:

- **Salience-Determined Bit Allocation** analyzes the structured distribution of weight salience to assign different precisions to groups of weights.

- **Salience-Weighted Quantizer Calibration** adjusts the quantizer parameters with a focus on the few highly salient weight elements within each weight group.

The experimental results show that SliM-LLM reduces perplexity compared to SOTA gradient-free PTQ methods while reducing memory usage by nearly 6x. The extended version with graidient-based optimization, SliM-LLM+, further improves the model performance.

## update after rebuttal
SliM-LLM achieved state-of-the-art performance in 2-bit quantization of large language models (LLMs) through two main contributions: Salience-Determined Bit Allocation and Salience-Weighted Quantization Calibration. While this approach effectively reduces the memory requirements of LLMs, it introduces a trade-off in terms of speedup — another critical benefit typically expected from quantization — in exchange for higher accuracy.
Overall, while the paper's claims are well-supported empirically and demonstrate strong experimental results and my other questions are well-addressed, further exploration is still needed regarding inference efficiency.

**Claims And Evidence:**

The analysis of the structuredness of global salience of weights is well supported by results across various layers and models. However, the presence of structural outliers in the output activations of LLMs—and the corresponding salient weight channels—has been extensively addressed in previous works.

The analysis of local salience is relatively underdeveloped, and it has also been explored in recent previous works [1][2].

[1] Yi, Ke, et al. "Rotated Runtime Smooth: Training-Free Activation Smoother for accurate INT4 inference." arXiv preprint arXiv:2409.20361 (2024).

[2] Yu, Mengxia, et al. "The super weight in large language models." arXiv preprint arXiv:2411.07191 (2024).

**Essential References Not Discussed:**

The authors may add references for works that mentions occurrence of unstructured outliers (salient values) such as [1], [2] of part 2.

**Experimental Designs Or Analyses:**

Inference efficiency is another important aspect just as quantized model performance. Although Table 5 presents comparison results with GPTQ, the main text lacks an explanation and analysis of the comparison scenario. The title of subsection 4.3, which claims to address efficient inference, appears to discuss a different topic.

**Methods And Evaluation Criteria:**

SBA and SQC are logically designed methods that directly address the aforementioned problems. Evaluating using perplexity and zero-shot tasks is appropriate, as these metrics are standard in LLM compression work.

**Other Comments Or Suggestions:**

I suggest to provide more details on

1) experiment settings on evaluating inference efficiency.

2) required time to apply proposed method

3) ablation study on group size

**Other Strengths And Weaknesses:**

**Strength**

Strong performance at 2-bit compared to baselines

**Weakness**

Lack of analysis on latency and throughput, Relatively bad performance on new models

**Questions For Authors:**

- In recent models like LLaMA-3, where there is a significant performance drop, can this method be considered practical?
- Why doesn't the paper provide results on the latest models for SliM-LLM+?

**Relation To Broader Scientific Literature:**

This research is influenced by existing work on post-training quantization of LLMs. The goal of preserving accuracy by effectively handling outlier or salient values was already established by previous studies, and this work extends that by demonstrating a more hardware-friendly implementation.

Additionally, by addressing both local and global salience, the paper represents a significant extension of the field.

**Theoretical Claims:**

The structured distribution of salience weights due to outlier activations has been pointed out and addressed in various works, and the existence of locally important weights also is not a novel issue introduced by this paper.

Rather than providing a mathematical proof, the paper seeks to validate these issues through experimental evidences.

---

> ### Author Rebuttal · Authors · 2025-04-01
>
> Dear Reviewer iFeq,
>
> Thank you for your valuable feedback and suggestions. We will address your questions and recommendations one by one.
>
> > Q1: (1)The analysis of local salience is relatively underdeveloped, and it has also been explored in recent previous works. (2)The authors may add references for works that mentions occurrence of unstructured outliers (salient values) such as [1], [2] of part 2.
>
> A: The variation in weight importance distribution is crucial for LLM quantization, particularly regarding local salience. As noted in prior studies, our key contribution is using the element sensitivity criterion (Definition 1) to analyze significant weight clustering and provide a theoretical explanation for salience clustering.
>
> We did not explore the underlying causes of local salience in detail, as they have been thoroughly discussed in works like [1] [2] and Section 3.3.2. Instead, we built on these insights to develop the Salience-Weighted Quantizer Calibration (SQC) strategy to reduce quantization errors. We appreciate your reference suggestions and will expand the discussion in the revised version.
>
> > Q2: Inference efficiency is another important aspect ... efficient inference, appears to discuss a different topic.
>
> Thank you for identifying the typo. This was a layout error, and we will correct it by including the full content of the "Efficient Inference on Device" section in the final version.
>
> Specifically, we extend the CUDA kernel in AutoGPTQ to support experimental mixed-precision inference (details in Appendix B.2). We evaluate LLaMA-7/13B and LLaMA-2-7B under 2/3-bit settings, showing that our approach maintains a high compression rate on GPUs while significantly improving model accuracy, with only a slight inference speed reduction on the A800 due to bit-width alignment. As 1-bit operations currently lack hardware support, additional storage and computation are required. We recognize the potential for further optimization in mixed-precision computing and aim to improve this in future work.
>
> > Comments Or Suggestions.
>
> A: (1)Thank you for your valuable suggestions! We will follow your advice and add the complete inference settings in Section 4.3.
>
> (2)The deployment time of SliM-LLM consists of two main parts: SBA bit-width search and SQC quantization parameter determination. As detailed in Section 3.2.2, the SBA dual-pointer search is highly efficient, and SQC computation time is comparable to standard calibration-based quantizers. SliM-LLM integrates seamlessly with existing PTQ strategies and, being training-free in a plug-and-play manner, completes 7B LLM quantization in about 25 minutes.
>
> (3)For the ablation study on group size, we have discussed the differences in detail in Appendix F and provided experimental results in Table 7. We will further highlight this content in the main text to improve the readability of the paper.
>
> > Questions For Authors.
>
> A: (1)Thank you for your insightful observations. You are correct about the challenges with models like LLaMA-3. Despite this, SliM-LLM maintains leading quantization performance and remains highly practical.
>
> As noted in prior works [1–4], quantization methods degrade more severely in knowledge-dense models like LLaMA-3, especially at ultra-low bit widths. Our experiments confirm this: in Table 1, LLaMA-3 8B with AWQ and GPTQ yields PPLs of 8.22 and 8.19 under 3-bit quantization, rising to 210 and 1.7e6 at 2-bit. This suggests that as models grow in knowledge density, conventional methods suffer greater quantization losses. In contrast, SliM-LLM achieves PPLs of 7.16 (3-bit) and 39.66 (2-bit), demonstrating its effectiveness.
>
> Additionally, we have included 4-bit quantization experiments, which are more practical, to further highlight SliM-LLM’s advantages on LLaMA-3.
>
> |Method|LLaMA-7B|LLaMA-13B|LLaMA2-7B|LLaMA2-13B|LLaMA3-8B|
> |-|-|-|-|-|-|
> |FP16|5.68|5.09|5.47|4.88|5.75|
> |AWQ|5.81|5.30|5.62|4.97|6.63|
> |GPTQ|5.85|5.20|5.61|4.98|6.50|
> |SliM-LLM|5.83|5.16|5.59|4.95|6.42|
>
> |Method|LLaMA-7B|LLaMA2-7B|
> |-|-|-|
> |Omniquant|5.77|5.58|
> |SliM-LLM+|5.75|5.57|
>
> (2)SliM-LLM+ incorporates our structured mixed-precision strategy into OmniQuant’s gradient quantizer. However, as OmniQuant and its comparator, AffineQuant, currently lack support for models like Gemma and Mixtral, we did not report results on these. We have contacted the OmniQuant authors to request updates and are actively working to extend SliM-LLM+ to more models.
>
> [1] Rotated Runtime Smooth: Training-Free Activation Smoother for accurate INT4 inference. arXiv:2409.20361.
>
> [2] Efficientqat: Efficient quantization-aware training for large language models[J]. arXiv:2407.11062.
>
> [3] Compressing large language models using low rank and low precision decomposition[J]. NeurIPS, 2024.
>
> [4] An empirical study of llama3 quantization: From llms to mllms[J]. Visual Intelligence, 2024.

---

### Official Review · Reviewer_Ut3n · 2025-03-13

**Overall Recommendation:** 4

**Summary:**

This paper introduces a group-wise mixed-precision quantization method for LLMs, addressing challenges in accuracy and efficiency. The key contributions are two strategies: SBA, which optimally allocates bit-widths by minimizing entropy divergence through Hessian and weight salience analysis, and SQC, which enhances salient weight representation by adjusting quantizer sensitivity using a calibration parameter. To handle group outliers, the method balances scale and zero-point adjustments with a focus on scale sharing. Both strategies are compatible with various quantizers. Experiments show that Slim-LLM outperforms existing methods on LLAMA and OPT models.

**Claims And Evidence:**

The submission's claimed phenomena and results are supported by detailed evidence.

**Essential References Not Discussed:**

N/A

**Experimental Designs Or Analyses:**

The authors provide detailed experiments demonstrating the performance advantages of SliM-LLM, particularly achieving better results than existing methods on the ppl evaluation metric and commonsense benchmarks. The appendix includes additional results on challenging tasks like math after quantization. Figure 5 presents thorough ablation studies showing the contribution of each component in the quantization method. The paper also provides real hardware deployment results for memory usage and inference speed, making the experiments comprehensive.

**Methods And Evaluation Criteria:**

The authors propose an innovative group-wise structured mixed-precision quantization strategy for LLMs, balancing accuracy and efficiency. It offers new insights into extreme compression under 2-bit and 3-bit settings and can be easily integrated into existing quantization tools. The observation and proof of the structured distribution of significant weights introduce a new paradigm for future LLM compression strategies.

**Other Comments Or Suggestions:**

There are minor typos, and the abbreviations WM and RM in Figure 5 lack annotations.

**Other Strengths And Weaknesses:**

Strengths:

1.	This work contributes a lot to the area of low bit quantization and compression. Make it efficient and accurate for different kinds of post training LLMs, especially under 2-bit and 3-bit. The structured mixed-precision proposed in SliM-LLM is a straightforward and effective method for the committee.

2.	Evaluation of the method’s efficiency is rigorous and supported by an extensive set of experiments. The results are well-documented and demonstrate the practical applicability and effectiveness of the proposed approach. The performance comparisons with existing methods highlight the strengths of the paper’s contributions, offering promising insights into its potential impact on the field. The inclusion of various evaluation metrics further strengthens the reliability and generalizability of the findings.

Weaknesses

1.	The authors are advised to consider testing on more challenging LLM benchmarks, such as GSM8K in the mathematics domain. Compared to MathQA, GSM8K may be more sensitive to the loss caused by quantization.

2.	In group-wise mixed-precision inference, the allocation of quantization scales across different channels and the process of dequantization could provide deeper insights into the inference details of this structure. The authors are encouraged to provide further explanations on these aspects.

**Questions For Authors:**

Although SLiM-LLM is a leading PTQ method in the 2-bit and 3-bit settings, which is a popular topic in the research field, it appears that the commonly used 4-bit quantization for industrial applications has not been explored in the paper. Could the authors provide a comparison of 4-bit PPL with RTN, AWQ, and GPTQ?

**Relation To Broader Scientific Literature:**

N/A

**Theoretical Claims:**

reviewed the correctness of the theoretical claims in Section 3.2.1. Figure 3 provides detailed evidence visualizing the weight clustering characteristics, and Section 3.2.1 theoretically establishes the relationship between the metric in Definition 3.1 and weight clustering. The discrete weight characteristics discussed in Section 3.2.2 are also supported by visualizations and additional explanations in the appendix. These proofs validate the proposed structured mixed-precision quantization framework.

---

> ### Author Rebuttal · Authors · 2025-04-01
>
> Dear Reviewer Ut3n,
>
> We sincerely appreciate your insightful feedback and suggestions. Below, we will respond to your questions and recommendations individually.
>
> > Q1: The authors are advised to consider testing on more challenging LLM benchmarks, such as GSM8K in the mathematics domain. Compared to MathQA, GSM8K may be more sensitive to the loss caused by quantization.
>
> A: Thank you for your suggestion. We have conducted additional tests to evaluate the performance of our method on more challenging datasets, such as GSM8K. The results are shown in Table below.
>
> |Model/Evaluation|Method|GSM8K|
> |-|-|-|
> |**LLaMA-7B**|GPTQ 3-bit|11.5|
> ||AWQ 3-bit|11.5|
> ||SliM-LLM 3-bit|11.5|
> |**LLaMA-7B**|GPTQ 2-bit|0.0|
> ||AWQ 2-bit|0.0|
> ||SliM-LLM 2-bit|9.2|
> |**LLaMA2-7B**|GPTQ 3-bit|13.0|
> ||AWQ 3-bit|13.4|
> ||SliM-LLM 3-bit|13.6|
> |**LLaMA2-7B**|GPTQ 2-bit|0.0|
> ||AWQ 2-bit|0.0|
> ||SliM-LLM 2-bit|10.3|
>
> The findings demonstrate that, even on GSM8K, our method exhibits less accuracy loss compared to other approaches, highlighting its robustness across different test sets.
>
> > Q2: In group-wise mixed-precision inference, the allocation of quantization scales across different channels and the process of dequantization could provide deeper insights into the inference details of this structure. The authors are encouraged to provide further explanations on these aspects.
>
> A: We have explained some details on how to allocate, store, and perform inference with quantization scales in Appendix B.2. This means that, while storing the quantized weights at extremely low bit widths, we also store the corresponding quantization scales for each row, and are able to perform dequantization operations on both scales and quantized integers on the CUDA Kernel, restoring them to the floating-point values required for inference. We will clarify and explicitly refer to this section in the main text.
>
> > Q3: There are minor typos, and the abbreviations WM and RM in Table 5 lack annotations.
>
> A: Thank you for your careful and thorough review! We will further check for any writing errors in the paper to ensure better readability. Regarding the issue you raised about the lack of explanations for WM and RM in Table 5, we would like to clarify that WM stands for Weight Memory and RM stands for Running Memory. We will also add the definitions of WM and RM in the relevant section to ensure they are accurately explained. Thank you for your helpful reminder.
>
> > Q4: Although SLiM-LLM is a leading PTQ method in the 2-bit and 3-bit settings, which is a popular topic in the research field, it appears that the commonly used 4-bit quantization for industrial applications has not been explored in the paper. Could the authors provide a comparison of 4-bit PPL with RTN, AWQ, and GPTQ?
>
> A: We sincerely appreciate your suggestion regarding comparative experiments with 4-bit quantization. We would like to clarify that although SliM-LLM is primarily optimized for 2-bit and 3-bit quantization, our method is also compatible with 4-bit mixed-precision quantization. In response to your suggestion, we have included additional experiments in the revised versions of Table 1 and Table 2. Furthermore, we have tested our method on several 4-bit models and are pleased to share the results with you below:
>
> |Method|LLaMA-7B|LLaMA-13B|LLaMA2-7B|LLaMA2-13B|LLaMA3-8B|
> |-|-|-|-|-|-|
> |FP16|5.68|5.09|5.47|4.88|5.75|
> |AWQ|5.81|5.30|5.62|4.97|6.63|
> |GPTQ|5.85|5.20|5.61|4.98|6.50|
> |SliM-LLM|5.83|5.16|5.59|4.95|6.42|
>
> |Method|LLaMA-7B|LLaMA2-7B|
> |-|-|-|
> |Omniquant|5.77|5.58|
> |SliM-LLM+|5.75|5.57|
>
> \*The results of RTN are too worse than GPTQ and AWQ to be listed here.

---

> > ### Comment · Reviewer_Ut3n · 2025-04-07
> >
> > Thank you for the further clarifications. The authors' additional explanations and insights have further strengthened the paper's contributions. The proposed Slim-LLM is well-validated and has clear practical relevance for LLM quantization and compression. I maintain my original accept rating.

---

> > > ### Author Response · Authors · 2025-04-07
> > >
> > > Dear Reviewer Ut3n,
> > >
> > > Thank you for your thoughtful and constructive feedback. We sincerely appreciate the time and effort you dedicated to reviewing our work. Your insights have been highly valuable in helping us identify areas for clarification and improvement. We have carefully considered your suggestions and incorporated them into our revised manuscript to enhance its clarity, rigor, and overall quality.
> > >
> > > Thank you again for your valuable input—it has been instrumental in strengthening our work.

---

### Official Review · Reviewer_BegC · 2025-03-16

**Overall Recommendation:** 3

**Summary:**

This paper proposes SliM-LLM, a post-training quantization (PTQ) framework for large language models (LLMs). Its core idea is to allocate bit-widths to weight groups adaptively and locally preserve important (salient) weights. The approach combines two techniques:
1. Salience-Determined Bit Allocation (SBA): Groups of weights are each assigned a suitable bit-width based on the group’s global importance (salience).
2. Salience-Weighted Quantizer Calibration (SQC): Within each group, a small subset of highly salient weights is given extra quantizer “attention” to reduce local discretization errors.

By focusing resources on channels or weight elements deemed most critical, SliM-LLM aims to achieve lower perplexity in ultra-low bit regimes (1–3 bits) without incurring major hardware overhead. Experimental results emphasize LLaMA (1, 2, 3) and OPT model families, showing that SliM-LLM outperforms existing PTQ baselines in perplexity reduction, and can be integrated into popular PTQ toolkits (e.g., GPTQ, OmniQuant). The paper also includes limited results on less-traditional architectures such as Gemma2 or Mixtral, in an effort to demonstrate some level of generalizability.

**Claims And Evidence:**

The paper makes strong claims that its proposed SliM-LLM method yields significant performance gains at very low bit-widths (1–3 bits) and generalizes effectively across large language models. While the results on LLaMA (and to a lesser extent OPT) support improved perplexity and moderate memory overhead, **some claims remain only partly substantiated**:

1. **Claim: “SliM-LLM balances compression rate and inference speed with limited overhead.”**
   - **Evidence**: The authors do show memory usage and token-throughput comparisons (e.g., Table 5), illustrating that at 2 bits, perplexity drops dramatically yet throughput also declines. The data partly confirms a trade-off.
   - **Gaps**: There is **no systematic exploration** of how severe the latency penalty can become in different scenarios (e.g., multi-GPU, different batch sizes). This means the paper has not comprehensively demonstrated that SliM-LLM offers a robust speed–accuracy trade-off across diverse deployment conditions.

2. **Claim: “SliM-LLM generalizes well beyond LLaMA/OPT.”**
   - **Evidence**: A few additional experiments (e.g., Gemma2-9B, Mixtral-8×7B) in the appendix are cited to show performance gains relative to GPTQ.
   - **Gaps**: Most of the core experiments still focus heavily on LLaMA/OPT families, leaving limited discussion or validation on how the method behaves for architectures with different normalization schemes, or for radically different designs (e.g., multimodal LLMs). The authors do not deeply explore how outlier channels are altered by extra normalization layers in Gemma2, nor whether the salience-based approach needs to be adapted.

3. **Claim: “SliM-LLM’s salience-based bit allocation and quantizer calibration handle local outliers effectively.”**
   - **Evidence**: The paper describes Salience-Weighted Quantizer Calibration (SQC) and provides ablation results indicating that it helps preserve a small fraction of crucial, high-magnitude weights.
   - **Gaps**: The experiments assume that outliers make up only around 1–5% of weights. For more extreme cases—where outlier or salient weights are more widespread—the paper does not offer specific evidence. Thus, it is unclear how robust the approach is if the model’s salience distribution does not cluster or if many channels simultaneously exhibit high salience.

4. **Claim: “SliM-LLM introduces only minor hidden overhead.”**
   - **Evidence**: The authors note a small increase in memory usage (e.g., ~0.1GB more than GPTQ at 2 bits on 7B-scale models) and mention that storing group-level bit-width maps is not a huge burden.
   - **Gaps**: There is no detailed breakdown of overhead for very large models (70B+ parameters) or a thorough demonstration that the additional groupwise metadata stays modest across scales. The paper would be stronger if it rigorously measured overhead as model size and group granularity vary.

Overall, while the **performance claims** at low-bit quantization (especially on LLaMA and OPT) are convincingly supported by perplexity gains and ablations, the **generalizability and overhead** claims are not explored in as much depth. Addressing the trade-offs with speed, broader model architectures, and more diverse salience patterns would make the evidence more conclusive.

**Essential References Not Discussed:**

None

**Experimental Designs Or Analyses:**

The experimental protocol mostly follows conventional PTQ setups (calibrate on a small set of input tokens, measure perplexity on standard test sets, compare with established baselines). The additional tables for Gemma2 and Mixtral do help, but only in a small-sample manner. Some analyses of memory overhead and speed trade-offs are provided (Table 5), but a deeper exploration of “how big is the hidden overhead for group-bit metadata?” or “which scenarios lose the most throughput from mixed-precision?” could be more thorough.

**Methods And Evaluation Criteria:**

The paper’s methodology—mixed-precision post-training quantization evaluated via perplexity on language modeling benchmarks—fits well with the core objective of aggressively compressing large transformer-based LLMs. Perplexity and zero-shot accuracy are reasonable metrics for confirming whether the quantized model still preserves core language understanding.

However, **some limitations remain**:

1. **Narrow Focus on LLaMA/OPT Families**
Most tests rely on LLaMA and OPT, which both use Pre-LN Transformer designs widely known for pronounced outlier channels. This raises the question of whether SliM-LLM’s promising results might partly hinge on the fact that these specific architectures emphasize salience clustering. If newer Transformers (or non-Transformer models) suppress or redistribute outliers—an effect sometimes seen with extra normalization layers —the gains from SliM-LLM might be less pronounced or require modifications. More diverse experiments on architectures without those strong outlier characteristics would clarify whether these gains arise purely from leveraging Pre-LN behaviors or can generalize more broadly.

*Sun, Mingjie, et al. "Massive activations in large language models." COLM2024.
*Kedia, Akhil, et al. "Transformers get stable: an end-to-end signal propagation theory for language models." ICML2024.
*Oh, Jaehoon, Seungjun Shin, and Dokwan Oh. "House of cards: Massive weights in llms." arXiv 2024.
*Sun, Wenfang, et al. "The Curse of Depth in Large Language Models." arXiv 2025.

2. **Sparse Analysis of Inference Trade-offs**
While the paper includes perplexity vs. token/s results, there is no deeper experiment varying batch sizes, GPU setups, or real-time conditions. It remains unclear whether SliM-LLM consistently meets real-world speed demands, especially under heavy concurrency or stricter latency targets. The discussion on throughput is largely single-GPU, and additional multi-GPU or distributed benchmarks could confirm if memory/performance scales similarly.

3. **Evaluation Datasets**
The authors rely on standard data like Wikitext2, C4, and limited zero-shot tasks. Although these are reasonable starting points, evaluating reasoning or multimodal tasks could uncover further performance nuances and better reflect practical deployments. While perplexity is a solid metric, real-world usage often involves more specialized tasks. The paper does not yet assess how SliM-LLM behaves under such domain-specific conditions.

Altogether, the presentation (PTQ for LLMs assessed by perplexity and memory/speed) is well-aligned with the basic goal of compressing large language models. Yet the limited architectural variety tested, the lack of detailed real-time inference experiments, and the narrow range of evaluation tasks leave open questions as to how broadly and reliably SliM-LLM can be applied in production-scale environments.

**Other Comments Or Suggestions:**

None

**Other Strengths And Weaknesses:**

**Strengths**

- Empirical gains at low bit quantization, often outperforming prior PTQ baselines.
- Fairly robust experiments on standard language modeling tasks.
- Engineering details (Appendix B) on how to pack bits group-wise, which is helpful for reproducibility.

**Weaknesses**

- Discussion of trade-offs in real-world inference scenarios remains somewhat limited. While the paper acknowledges that speed can drop at 2 bits, it does not systematically measure it across different GPU kernels or larger-batch conditions.
- The usage of only a few beyond-LLaMA architectures (Gemma2, Mixtral) is a first step, but the explanation for why it should work in every scenario is short.
- Overall paper formatting (some references, layout) does not strictly follow typical ICML style, giving an unfinished impression.
- The theoretical discussion is primarily heuristic—though typical in quantization research, it might have benefited from deeper analysis or direct ablation on truly scattered salience distributions.

**Questions For Authors:**

- Could the authors elaborate how SQC would respond if *every* channel had moderate outliers rather than a few large outlier channels? Would the search for τ-parameters become unstable or scale poorly?
- Would additional codebook-based or vector-quantization steps (e.g., SpQR) further improve results?

- The paper includes a small table (Table 5) comparing memory usage and token throughput. Could the authors detail how much extra overhead is stored in the group-wise bit metadata at 2 bits vs. 3 bits, especially for large models (e.g., 70B+)?
- Do you foresee more advanced GPU kernels that specifically accelerate group-wise mixed-precision to mitigate the 2-bit throughput drop?

- If future LLM variants (e.g., multi-modal or heavily fine-tuned) produce qualitatively different activation distributions, how stable is the salience-based approach?
- Are additional calibration samples or adaptive re-quantization steps needed to maintain performance?

**Relation To Broader Scientific Literature:**

- The paper aligns with ongoing efforts to aggressively compress LLMs under 3-bit (e.g., QuIP, OmniQuant, GPTQ, etc.).
- Extends the salience-based approach of GPTQ but adds a structured mixed-precision angle, reminiscent of older “HAWQ” or “APTQ” but at a finer group level.
- Prior works that address outlier channels or element-wise mixed precision are referenced (e.g., AWQ).

**Theoretical Claims:**

The authors briefly justify salience clustering by referencing Hessian approximations and outlier activation channels. These are not extremely formal proofs but rather high-level derivations consistent with prior work on Hessian-based importance. No obvious flaws stand out, though they remain somewhat heuristic in nature.

---

> ### Author Rebuttal · Authors · 2025-04-01
>
> Dear Reviewer BegC,
>
> Thank you for your valuable feedback. We have summarized your questions and concerns. Due to the character limit for reply, if there are any question that are not detailed, we will further reply in next stage. Thanks you so much.
>
> > Q1:(1) There is...diverse deployment conditions. (2)Discussion of trade-offs...conditions.
>
> As shown in Table 5, we analyzed deployment speed under different memory compression levels using a batch size of 1 (align with other works), focusing on extreme compression for single-GPU. We added inference experiments for the compressed LLaMA-7B model on different GPUs, testing various batch sizes.
> |GPU|Method|Bit|Batch|Token/s|
> |-------------|------------|-------|---------|-----------|
> |RTX4090|SliM-LLM|2|4|55.1|
> ||||8|34.4|
> |A100|SliM-LLM|2|4|59.2|
> ||||8|40.2|
>
> Slight speed decreases with larger batch, consistent across frameworks like AutoGPTQ and AutoAWQ.
>
> > Q2: (1)Most of the core...to be adapted. (2)The usage of...is short. (3) If future LLM...approach?
>
> The LLaMA and OPT series are among the most widely downloaded and applied LLM architectures in the community. They are also commonly used as base models in other works. As you noted, we include more experiments in Table 11 to evaluate models with extra normalization layers. We claim that salience is a relative concept, as LLM training inherently produces such salience[1][2], even in multimodal or fine-tuned models. While architectures like Gemma2 use extra normalization layers, relative salience of features still emerges.
>
> SliM-LLM focuses on compressing LLMs. Follow your suggestion, we expanded multimodal task on the LLaVA-Next 8B model(N:collapse accuracy).
> ||#W|#G|AI2D|ChartQA|DocVQA|MMBench|
> |--------------------------|------|-------|--------|------------|-----------|------------|
> |GPTQ|3|128|66.2|65.1|75.6|67.4|
> ||2|128|N|N|N|N|
> |AWQ|3|128|67.7|65.4|74.4|68.0|
> ||2|128|N|N|N|N|
> |SliM-LLM|3|128|68.2|67.5|74.8|68.9|
> ||2|128|57.2|49.3|60.6|60.9|
>
> [1]From Attention to Activation: Unraveling the Enigmas of Large Language Models.
>
> > Q3: (1)The experiments a...exhibit high salience. (2) The theoretical discussion...distributions.
>
> The proportion of local outlier weights within groups is not an assumption but an observation supported by prior studies. Section 3.2.2 and previous work[1] consistently show that salient weights within groups remain a small proportion. Theorem 1 and detailed proofs in Appendix G explain this phenomenon.
>
> [1] SpQR: A sparse-quantized representation for near-lossless LLM weight compression.
>
> > Q4: (1)There is no detailed breakdown ...vary. (2)The paper...large models (e.g., 70B+)?
>
> In Table 7, we provided the quantization performance of four LLMs under different group sizes. SliM-LLM introduces negligible storage overhead, which decreases further as model size grows. For instance, with LLaMA2-70B (group sizes = 64, 128, 256), a single transformer layer (size 8192 × 8192) requires a group matrix of size 8192 × 64/128/256. Using three precision levels, only a 2-bit flag (e.g., 01 = 1-bit, 10 = 2-bit, 11 = 3-bit) is needed per group. The additional storage overhead is $\frac{2}{8192 \times 64/128/256}$—virtually negligible at scale. Other quantization parameters are identical to frameworks like GPTQ.
>
> Based on your suggestion, we will add 70B models results.
> |LLaMA-2-70B #W|Method|WM|RM|PPL|Token/s|
> |------------------|------------|---------|----------|-----------------|-----------|
> |3-bit|GPTQ|28.0G|34.9G|3.85|6.5|
> ||SliM-LLM|28.0G|35.2G|3.67|6.2|
> |2-bit|GPTQ|16.4G|23.3G|8.78|9.7|
> ||SliM-LLM|16.5G|23.5G|6.28|8.4|
>
> > Q5: The authors rely on standard...domain-specific conditions.
>
> Our paper evaluates quantized model performance on 13 benchmarks, including Wikitext2, C4, and 8 zero-shot tasks, aligning with other LLM quantization works. Appendix Table 13 includes benchmarks from Humanities, Social Sciences, and STEM, with reasoning tasks like MathQA highlighting SliM-LLM's strengths in math reasoning. We will add results for LLaVA-Next-8B on 4 multimodal tasks in Q2.
>
> >Questions For Authors:
>
> (1)As shown in Definition 3.1 and Theorem 1, salience is influenced by weight magnitudes and the Hessian. If all weights exhibit similar salience, the distinction between salient and non-salient weights (Equation 5) becomes negligible. In such cases, τ converges to 1, simplifying quantization and allowing parameters to be derived from basic statistics.
>
> (2)Codebook-based methods can improve accuracy but significantly reduce inference speed—e.g., 2-bit code-book can be 3× slower. SliM-LLM’s structured mixed-precision quantization achieves competitive accuracy while maintaining practical inference speeds.
>
> (3)Frameworks like recent HQQ show promising advancements in GPU kernels for 2-bit quantization. These developments could further improve SliM-LLM's inference speed.
>
> (4)For calibration, SliM-LLM used only 128 random samples from WikiText2, avoiding additional data or adaptive re-quantization.

---

> > ### Comment · Reviewer_BegC · 2025-04-06
> >
> > I appreciate the authors for the detailed responses and additional experimental results. Although some of the answers do not sufficiently answer the questions, I believe the additional experimental results, as well as the in-depth discussion, would be valuable for future research in this field. Therefore, I raise my original rating.

---

> > > ### Author Response · Authors · 2025-04-07
> > >
> > > Dear Reviewer BegC,
> > >
> > > Thank you sincerely for your detailed review and for the generous score adjustment. We greatly appreciate your recognition of our work and your thoughtful suggestions, which have been instrumental in further refining and strengthening our manuscript.
> > >
> > > In response to your insightful comments, we have conducted additional experiments to substantiate our claims and further assess the robustness of our proposed method. These new results—covering various batch size and GPU settings, multimodal tasks, and the performance of larger-scale models—provide stronger empirical support and a deeper understanding of our method's effectiveness.
> > >
> > > We will incorporate these supplementary results into the final version of the paper to enhance its completeness, rigor, and credibility. Once again, thank you for your kind support and invaluable feedback.

---

### Decision · Program_Chairs · 2025-05-01

**Decision:**

Accept (poster)

**Comment:**

The paper makes a solid contribution to LLM quantization, presenting a novel and effective PTQ method (SliM-LLM) that achieves state-of-the-art results, particularly in the challenging ultra-low-bit regime (2/3-bit). The core ideas are well-motivated, and the empirical validation, strengthened significantly by the rebuttal, is convincing. While the analysis of inference speed trade-offs could be more exhaustive, the demonstrated accuracy gains at high compression rates are significant.

The authors' engagement during the rebuttal phase substantially improved the paper and addressed many key reviewer concerns. Given the strong results, methodological contribution, and positive reception (3 Weak Accepts and 1 Accept, with two scores raised post-rebuttal), the paper warrants acceptance. The authors are encouraged to incorporate the additional results and discussions from the rebuttal, particularly regarding inference efficiency trade-offs and careful positioning relative to prior work on salience, into the final version.